# Probabilistic Performance Guarantees for Multi-Task Reinforcement Learning

Yannik Schnitzer [* 1]   Mathias Jackermeier [* 1]   Alessandro Abate [1]   David Parker [1]

## Abstract

Multi-task reinforcement learning trains generalist policies that can execute multiple tasks. While recent years have seen significant progress, existing approaches rarely provide formal performance guarantees, which are indispensable when deploying policies in safety-critical settings. We present an approach for computing high-confidence guarantees on the performance of a multi-task policy on tasks not seen during training. Concretely, we introduce a new generalisation bound that composes (i) per-task lower confidence bounds from finitely many rollouts with (ii) task-level generalisation from finitely many sampled tasks, yielding a high-confidence guarantee for new tasks drawn from the same arbitrary and unknown distribution. Across state-of-the-art multi-task RL methods, we show that the guarantees are theoretically sound and informative at realistic sample sizes.

## 1. Introduction

Multi-task reinforcement learning (MTRL) aims to train a single policy that performs reliably across many environments, instead of learning a separate policy for each one. Tasks encountered during training and deployment are modelled as draws from an underlying task distribution, which is generally unknown and need not satisfy any parametric assumptions. Here, tasks correspond to entire environments and may differ in dynamics, reward structure, objectives, and initial conditions. The training objective then depends on the use case: one may target high average performance across the distribution (Parisotto et al., 2016; Schaul et al., 2015), or adopt risk-aware objectives that prioritise reliability on difficult tasks and limit severe failures (Teh et al., 2017; Collins et al., 2020; Greenberg et al., 2023).

Deploying a multi-task policy in safety-critical settings calls for quantifying how the policy will behave on a new task that was not encountered during training, i.e., under zero-shot task generalisation (Oh et al., 2017). This is crucial, for instance, in autonomous driving, where weather and traffic conditions vary widely, or in robotic manipulation, where changes in objects, friction, and contact dynamics can lead to failures. In such settings, we require principled guarantees certifying that a policy meets a required performance level with high confidence also on unseen tasks.

This paper addresses that certification problem. We provide guarantees for any learned multi-task policy, regardless of the training algorithm. Given a set of tasks sampled from the unknown task distribution and a finite number of rollouts of the policy on each sampled task, we derive a high-confidence guarantee for performance on the next, previously unseen task. For a user-chosen performance requirement, the guarantee provides a lower bound on the probability that the policy meets this requirement on a new task. This makes the result directly interpretable as a safety certificate.

The key technical challenge is that certification must account for two sources of uncertainty. First, we observe only finitely many tasks, which provide limited information about the unknown task distribution over potentially uncountably many tasks. Second, on a sampled task, the policy's true performance is typically not available in closed form and must be estimated from a finite number of rollouts.

We develop a certification method that explicitly propagates rollout uncertainty through task-level generalisation. Our analysis proceeds in two stages: we first construct sound lower confidence bounds on performance on each sampled task from finite rollouts, and then aggregate these probabilistic bounds into a single certificate that generalises to unseen tasks from the same distribution. The resulting certification procedure is computationally lightweight and applies to any multi-task or meta-reinforcement learning pipeline, since it leverages only observed rollout data.

We evaluate the method across state-of-the-art training algorithms. The resulting certificates are statistically sound and, importantly, informative at practical sample sizes, providing actionable guarantees without requiring extreme numbers of tasks or rollouts. Moreover, the required number of samples depends on user-chosen confidence rather than on the dimensionality of the state or action spaces, making the approach applicable to high-dimensional and complex domains.

---

[*]Equal contribution   [1]University of Oxford, UK. Correspondence to: Yannik Schnitzer <yannik.schnitzer@cs.ox.ac.uk>.

*Proceedings of the $43^{rd}$ International Conference on Machine Learning*, Seoul, South Korea. PMLR 306, 2026.

## 2. Related Work

**Multi-task and meta reinforcement learning.** MTRL aims to learn a single policy that performs well across a family of related tasks. Classic MTRL and generalisation-oriented formulations include Actor-Mimic (Parisotto et al., 2016), universal value function approximators (Schaul et al., 2015), and goal-conditioned policies (Oh et al., 2017). Closely related is *meta*-RL, where the objective is fast adaptation. The agent is trained so that it can quickly adapt to a new task using a limited amount of task-specific experience, for example via gradient-based meta-learning (Finn et al., 2017), recurrent or memory-based adaptation (Duan et al., 2016; Wang et al., 2017), or latent-variable approaches that infer task embeddings from context (Rakelly et al., 2019). Beyond average-case performance, several works study robust or risk-aware variants that optimise for reliability on difficult tasks and reduce severe failures (Teh et al., 2017; Collins et al., 2020; Greenberg et al., 2023). Our work is complementary: we do not propose a new MTRL or meta-RL algorithm, but provide principled post-training, rollout-based certificates for any learned policy.

**Single-task guarantees and distribution-free evaluation.** High-confidence guarantees for a fixed task are standard in learning theory and typically follow from concentration inequalities, e.g., via PAC-style generalisation bounds (Langford, 2005; Boucheron et al., 2013). In RL, related tools support rollout-based evaluation with statistical guarantees (Agha & Palmskog, 2018; Budde et al., 2025). These methods provide guarantees for performance within a fixed task, where the only uncertainty arises from finite rollout sampling. In the multi-task setting, we must additionally generalise from finitely many sampled tasks to an unknown task distribution, while performance within each sampled task is itself only estimated from rollouts. Our bound jointly accounts for both sources of uncertainty.

**Certificates via learned finite-state models.** Schnitzer et al. (2025) study performance guarantees across an unknown distribution over tasks in a setting where tasks share a fixed finite-state structure and differ only in unknown transition probabilities. Their approach builds uncertainty sets over transition dynamics and derives guarantees by reasoning about a model-based over-approximation of the system. This is well-suited to finite-state models with known structure, but it ties the certificate to the complexity of the constructed model and its uncertainty representation. By contrast, we treat tasks as arbitrary MDPs with potentially different reward structures and continuous state spaces, and we certify performance directly from rollouts of the learned policy. Our certificates therefore do not require a model to be built and do not degrade with state-space dimension.

**MTRL generalisation bounds.** The work of Kostas et al. (2021) derives distribution-level guarantees for policy performance on unseen tasks. A key distinction is that their analysis assumes access to per-task performances in closed form, without estimation error. In realistic MTRL settings, however, per-task performance is typically not available in closed form and must be estimated from rollouts, which introduces an additional source of uncertainty that affects the validity of distribution-level guarantees. Our contribution is to make this estimation layer explicit and propagate it through the task-level generalisation step: we aggregate rollout-based *probabilistic* lower bounds on per-task performance, rather than treating per-task performance as known exactly. This yields a sound certificate under the data regime encountered in complex and continuous-control domains.

## 3. Background

We begin by introducing some notations, the mathematical foundations of RL, and the concept of MTRL.

**Reinforcement learning.** We model RL environments as *Markov decision processes* (MDPs) (Puterman, 1994). An MDP is a tuple $\mathcal{M} = (\mathcal{S}, \mathcal{A}, \mathcal{T}, \rho, r, \gamma, T)$, where $\mathcal{S}$ is the state space, $\mathcal{A}$ is the action space, $\mathcal{T} \colon \mathcal{S} \times \mathcal{A} \to \Delta(\mathcal{S})$ is the transition function, $\rho \in \Delta(\mathcal{S})$ is the initial state distribution, $r \colon \mathcal{S} \times \mathcal{A} \times \mathcal{S} \to \mathbb{R}$ is the reward function, $\gamma \in [0,1)$ is the discount factor, and $T \in \mathbb{N} \cup \{\infty\}$ is the time-horizon. We denote by $\mathcal{T}(s' \mid s, a)$ the probability of transitioning to state $s'$ from state $s$ under action $a$. A (memoryless) *policy* $\pi \colon \mathcal{S} \to \Delta(\mathcal{A})$ maps states to distributions over actions and induces a distribution over trajectories $\tau = (s_0, a_0, r_0; s_1, a_1, r_1; \ldots)$ in $\mathcal{M}$, where $s_0 \sim \rho$, $a_t \sim \pi(\cdot \mid s_t)$, $s_{t+1} \sim \mathcal{T}(\cdot \mid s_t, a_t)$, and $r_t = r(s_t, a_t, s_{t+1})$. We write $\tau \sim \mathcal{M}^\pi$ for trajectories generated according to this distribution. The standard reinforcement learning problem is to find a policy that maximizes the *expected discounted return* (Sutton & Barto, 2018):

$$J_{\mathcal{M}}(\pi) = \mathop{\mathbb{E}}_{\tau \sim \mathcal{M}^\pi} \left[ \sum_{t=0}^{T} \gamma^t r_t \right].$$

**Multi-task reinforcement learning.** In MTRL, we consider a distribution $\mathcal{D}$ over MDPs (the *tasks*). The goal is to learn a single policy that performs *well* across the distribution of tasks, up to some notion of good performance. The distribution can be arbitrary, it does not need to belong to any standard parametric family and may be *entirely unknown* to the agent and the user.

The traditional formulation of MTRL seeks a policy $\pi_{\mathbb{E}}^*$ that maximises the expected return over tasks:

$$\pi_{\mathbb{E}}^* = \arg\max_{\pi} \mathop{\mathbb{E}}_{\mathcal{M} \sim \mathcal{D}} [J_{\mathcal{M}}(\pi)].$$

However, a policy optimising this objective may not perform uniformly well across all tasks and, while performing well on average, may underperform on some difficult tasks. To account for worst-case performance, robust MTRL considers worst-case or risk-sensitive objectives (Teh et al., 2017; Collins et al., 2020). For instance, recent work (Greenberg et al., 2023) optimises the *conditional value at risk* (CVaR) of performance over the task distribution. Our framework equally handles expectation-based, CVaR-based and true worst-case training objectives.

**Performance metrics.** While the expected discounted return is the canonical performance metric in RL, our framework allows a broader class of metrics. We distinguish finite- and infinite-horizon settings depending on whether $T < \infty$ or $T = \infty$. Let $\Omega := (\mathcal{S} \times \mathcal{A} \times \mathbb{R})^\omega$ and $\Omega^* := (\mathcal{S} \times \mathcal{A} \times \mathbb{R})^*$ denote the spaces of infinite and finite trajectories.

At a high level, our goal is to obtain, with high probability, a lower bound on $J_\mathcal{M}(\pi)$ for a policy $\pi$ when executed on a new, unseen task $\mathcal{M} \sim \mathcal{D}$. We will do so using a collection of sampled tasks, and for each sampled task we will estimate $J_\mathcal{M}(\pi)$ from rollouts of policy $\pi$. For finite-horizon problems ($T < \infty$), the trajectory-level performance is fully observable from a length-$T$ rollout, so we place no additional restrictions on the metric. For infinite-horizon problems ($T = \infty$), by contrast, we only ever observe a finite prefix of an infinite trajectory. To connect what we can observe to the infinite-horizon quantity we are interested in, we assume that the metric evaluated on prefixes provides a lower bound on the metric of the full trajectory.

Formally, a *trajectory metric* is a map $\mathcal{G}: \Omega \to \mathbb{R}$. We assume $\mathcal{G}$ admits a finite-prefix version $\mathcal{G}^*: \Omega^* \to \mathbb{R}$ such that $\mathcal{G}^*(\tau[..h])$ depends only on the prefix $\tau[..h] = (s_0, a_0, r_0; \dots; s_h, a_h, r_h)$ and can be computed after observing the first $h$ steps. The induced *performance metric* is the expected value over trajectories:

$$J_\mathcal{M}(\pi) := \mathop{\mathbb{E}}_{\tau \sim \mathcal{M}^\pi} [\mathcal{G}(\tau)].$$

We restrict to *monotone* trajectory metrics: $\mathcal{G}$ is monotone if

$$\mathcal{G}^*(\tau[..h]) \leq \mathcal{G}(\tau) \qquad \text{for all } \tau \in \Omega, \ h \in \mathbb{N}.$$

In other words, observing more of a trajectory cannot decrease its measured performance, so any finite-prefix evaluation is a valid lower bound on the value of the full, infinite trajectory. This condition holds, for example, for discounted return with nonnegative rewards, with $\mathcal{G}^*$ being the truncated discounted sum, and for sparse success criteria common in robotics: if $\mathcal{G}(\tau) \in \{0, 1\}$ indicates whether $\tau$ ever reaches some target set of states, then $J_\mathcal{M}(\pi)$ is exactly the success probability under $\pi$. Finally, monotonicity is a per-trajectory property: it is separate from the inherent stochastic variability across rollouts, which we will formally quantify when estimating $J_\mathcal{M}(\pi)$ from sample rollouts.

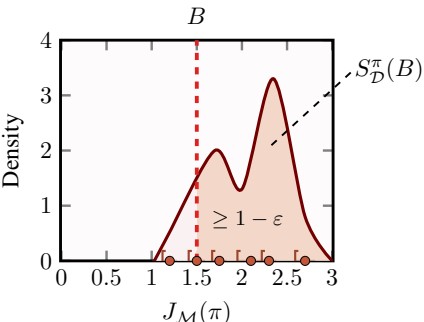

*Figure 1.* Example distribution of performances $J_\mathcal{M}(\pi)$ induced by a task distribution $\mathcal{D}$. Using sampled tasks and rollout-based lower confidence bounds (red brackets) around empirical per-task performance estimates (red dots), our results certify a lower bound on the safety level $S^\pi_\mathcal{D}(B) \geq 1 - \varepsilon$ for a user-chosen threshold $B$.

**Necessity and relaxation of monotonicity.** While monotonicity may seem a strong assumption, we highlight that it is in fact the minimal required assumption for arbitrary trajectory metrics in the infinite horizon setting. Without this assumption, a trajectory could trigger an unbounded degradation in performance after any observed prefix, making it impossible to certify a lower bound on $\mathcal{G}(\tau)$ from finite observations alone.

Furthermore, we note that many standard RL objectives can be readily transformed into monotone metrics. For instance, consider the infinite-horizon discounted return with rewards bounded from below by some constant $r_{\min}$, which is not monotonic for $r_{\min} < 0$. We can recover a monotone metric with one of the two following simple strategies:

1. *Worst-case geometric adjustment.* If $r_{\min}$ is known, we can account for the maximum possible future loss by redefining the finite-prefix statistic as

$$\mathcal{G}^*(\tau[..h]) := \sum_{t=0}^{h} \gamma^t r_t + \frac{\gamma^{h+1}}{1-\gamma} r_{\min},$$

which deducts the maximum possible future loss incurred beyond step $h$. This adjusted statistic converges to the true return from below as $h \to \infty$, thereby satisfying the monotonicity requirement of our framework.

2. *Reward shifting.* Alternatively, one can shift all rewards by $|r_{\min}|$ to render them non-negative, which preserves the relative performance ordering of policies (Ng et al., 1999). Since the resulting rewards are all non-negative, our framework can be readily applied.

## 4. Problem Setting

Our goal is to provide *high-confidence* guarantees on the performance of a learned MTRL policy on a new, unseen

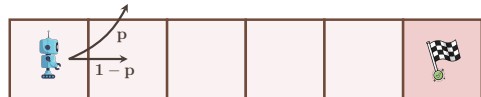

*(a)* Agent navigation domain with slip probability $p$.

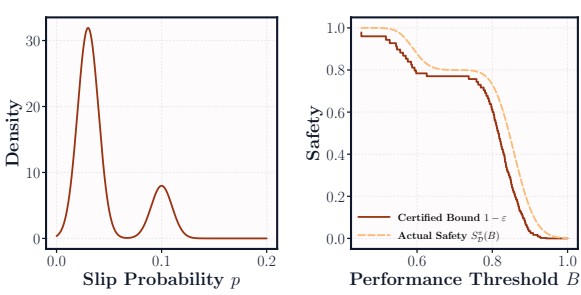

*(b)* Task distribution $\mathcal{D}$.  *(c)* Safety and certified bound.

*Figure 2.* Navigation example where the slip probability $p$ defines the task. Panel 2c reports the true safety level $S_{\mathcal{D}}^{\pi}(B)$ (dashed) and the certified lower bound $1 - \varepsilon$ (solid curve) from Theorem 5.1 at confidence $1 - \delta = 0.99$ for any performance threshold $B$.

task drawn from the task distribution $\mathcal{D}$. Importantly, our guarantees are *algorithm-agnostic*: we take the policy $\pi$ as given and do not assume anything about how it was trained. We assume access to $n$ i.i.d. sample tasks $\mathbb{M} := \{\mathcal{M}_i \sim \mathcal{D}\}_{i=1}^n$, and, for each task $\mathcal{M}_i$, a set of $m_i$ i.i.d. rollout trajectories collected by executing $\pi$ in $\mathcal{M}_i$, $\mathbb{T}_i := \{\tau_{i,j} \sim \mathcal{M}_i^{\pi}\}_{j=1}^{m_i}$. Intuitively, we evaluate $\pi$ on multiple tasks from the same distribution and use the resulting rollouts to *certify* how likely $\pi$ is to achieve a desired performance level on the next unseen task sampled from the same distribution.

**Definition 4.1** (Safety). Fix a performance threshold $B \in \mathbb{R}$. The *safety* of $\pi$ at level $B$ under $\mathcal{D}$ is

$$S_{\mathcal{D}}^{\pi}(B) := \Pr_{\mathcal{M}\sim\mathcal{D}}\left[J_{\mathcal{M}}(\pi) \geq B\right]. \tag{1}$$

Equivalently, the *risk* of falling below $B$ is $1 - S_{\mathcal{D}}^{\pi}(B)$, i.e., the CDF of random variable $J_{\mathcal{M}}(\pi)$ at threshold $B$.

The threshold $B$ is chosen by the user to reflect the desired level of performance in a given domain or application, for instance, a required success probability, or a minimum acceptable return or resource-efficiency level.

In practice, since the task distribution $\mathcal{D}$ is unknown, the safety level $S_{\mathcal{D}}^{\pi}(B)$ must be inferred from finite data $\{\mathbb{M}, (\mathbb{T}_i)_{i=1}^n\}$. This entails two sources of randomness: (i) the sampled tasks $\mathcal{M}_i \sim \mathcal{D}$, and (ii) the rollout randomness within each task, $\tau_{i,j} \sim \mathcal{M}_i^{\pi}$. Accordingly, we will construct a data-dependent bound $\varepsilon$ such that, for any user-chosen confidence level $1 - \delta$ with $\delta \in (0, 1)$,

$$\Pr\left[S_{\mathcal{D}}^{\pi}(B) \geq 1 - \varepsilon\right] \geq 1 - \delta, \tag{2}$$

where the outer probability is over both the draw of sample tasks $\mathbb{M}$ and the rollouts $\mathbb{T}_1, \ldots, \mathbb{T}_n$. In other words, with

probability at least $1 - \delta$ over the collected data, the true probability $S_{\mathcal{D}}^{\pi}(B)$ that $\pi$ achieves performance at least $B$ on an unseen task is at least $1 - \varepsilon$. For any given performance threshold $B$, we aim to certify a safety level $1 - \varepsilon$ that is as large as possible, while meeting the user-chosen confidence requirement $1 - \delta$, given available sample data.

Guarantees of the form (2) are *probably approximately correct* (PAC) (Valiant, 1984): *probably* refers to the confidence level $1 - \delta$ under finite samples, and *approximately* refers to the residual risk $\varepsilon$ quantified by the certificate.

Figure 1 illustrates the intuition. For a policy $\pi$, the task performance $J_{\mathcal{M}}(\pi)$ is a random variable induced by $\mathcal{M} \sim \mathcal{D}$, an unknown task distribution. From the finite samples $\mathcal{M}_i$ in $\mathbb{M}$ and corresponding rollouts $\mathbb{T}_i$, we will derive a certificate $(B, \varepsilon, \delta)$ of the form (2) for the performance of $\pi$ on any newly sampled, unseen task.

The main technical challenge is to account for *both* sources of uncertainty: (i) the fact that we only observe a finite set of tasks sampled from the unknown distribution $\mathcal{D}$, and (ii) the fact that, within each sampled task $\mathcal{M}_i$, the performance $J_{\mathcal{M}_i}(\pi)$ must itself be estimated from finitely many rollouts. If the task-level performances $J_{\mathcal{M}_i}(\pi)$ were known exactly, as assumed in Kostas et al. (2021), the problem reduces to a standard estimation task over i.i.d. samples. In realistic settings, however, the dynamics are typically continuous or otherwise unavailable in closed form, so $J_{\mathcal{M}_i}(\pi)$ cannot be computed analytically and must be approximated from data. Concretely, we only observe an empirical estimate $\hat{J}_{\mathcal{M}_i}(\pi)$ constructed from the rollout set $\mathbb{T}_i$.

We develop guarantees that *jointly* account for task-sampling uncertainty and rollout-estimation uncertainty. The resulting certificates are both statistically valid and tight in practice.

### 4.1. Illustrative Example

Consider the grid world in Figure 2a. The agent starts in the leftmost cell and receives reward 1 upon reaching the rightmost cell; all other rewards are 0. At each step, the episode may terminate due to a slip with probability $p \in [0, 1)$. The optimal policy $\pi$ is to move to the right at every step. We can view different slip probabilities $p$ as different tasks (different environment dynamics), i.e., each task $\mathcal{M}$ is parameterised by its slip probability $p$.

In this domain, the performance $J_{\mathcal{M}}(\pi)$ is the success probability of reaching the goal without slipping off track. Since the agent must survive 5 steps to reach the goal, we have $J_{\mathcal{M}}(\pi) = (1 - p)^5$. As an example task distribution, we consider the Gaussian mixture over $p$ shown in Figure 2b. Because the distribution is known in this synthetic example, we can accurately compute its CDF and thus the *true* safety

$$S_{\mathcal{D}}^{\pi}(B) = \Pr_{p\sim\mathcal{D}}\left[(1 - p)^5 \geq B\right],$$

i.e., the probability mass of slip probabilities $p$ for which the policy achieves performance at least $B$. The resulting true safety curve is shown as dashed lines in Figure 2c.

Figure 2c also shows, as a solid line, the *certified* safety bounds $1 - \varepsilon$ produced by our method at confidence level $1 - \delta = 0.99$. The certificate is computed from $n = 250$ sampled tasks $p \sim \mathcal{D}$ and $m_i = 10^3$ rollouts per task. As we shall argue later, the certification procedure and its sample requirements do not scale with the complexity of the domain. Accordingly, a similar number tasks and rollouts yields comparably tight certificates in high-dimensional continuous-control domains, as we demonstrate in Section 6.

This illustrative domain highlights the two layers of uncertainty we address: we only observe finitely many tasks from the unknown distribution over $p$, and within each task the performance $(1 - p)^5$ is not observed directly but must be estimated from finitely many rollouts. Since the true safety is explicitly computable in this toy setting, the figure allows a direct comparison between the certified bound and the ground truth, showing that the certificate is sound and tight.

## 5. Computing Performance Certificates

Our technique for high-confidence guarantees on the performance of $\pi$ in an unseen task addresses the two layers of uncertainty in MTRL: we only observe finitely many tasks from the unknown distribution $\mathcal{D}$, and we only observe finitely many rollouts in each task. Accordingly, our analysis consists of two steps. First, we use rollouts to construct *sound* high-confidence lower bounds on the performance $J_{\mathcal{M}_i}(\pi)$ in each sampled task $\mathcal{M}_i$. Then, we present a new result that turns these per-task bounds into a single, general guarantee for any next unseen task $\mathcal{M} \sim \mathcal{D}$. The key contribution is to propagate rollout uncertainty through the task-level generalisation step: we aggregate *probabilistic* lower bounds on $J_{\mathcal{M}_i}(\pi)$ obtained from the rollouts rather than assuming the task performances are known exactly.

### 5.1. Bounding Per-Task Performance

We first discuss how to obtain high-confidence lower bounds on the performance $J_{\mathcal{M}}(\pi) = \mathbb{E}_{\tau \sim \mathcal{M}^\pi}[\mathcal{G}(\tau)]$ of a policy $\pi$ in a task $\mathcal{M}$ from a finite sample set of $m$ i.i.d. rollouts $\mathbb{T} = \{\tau_j \sim \mathcal{M}^\pi\}_{j=1}^{m_i}$. We define the observable per-rollout statistic $X_j := \mathcal{G}^*(\tau_j)$, which evaluates the performance on the (finite) observed rollout, and the empirical performance $\widehat{J}_{\mathcal{M}}(\pi) := \frac{1}{m} \sum_{j=1}^{m} X_j$.

Notice that, when $T = \infty$, $J_{\mathcal{M}}(\pi)$ depends on infinite trajectories and cannot be observed directly from finite-length rollouts. Since we assume performance to be monotone, i.e. $\mathcal{G}^*(\tau[..h]) \leq \mathcal{G}(\tau)$ for all $\tau$ and $h \in \mathbb{N}$, we have

$$\mathbb{E}[X_j] = \mathbb{E}[\mathcal{G}^*(\tau[..h])] \leq \mathbb{E}[\mathcal{G}(\tau)] = J_{\mathcal{M}}(\pi),$$

so any *lower* bound on $\mathbb{E}[X_j]$ is also a valid lower bound on $J_{\mathcal{M}}(\pi)$. Therefore, monotonicity allows us to certify infinite-horizon performance from finite rollouts and the observable statistics $X_1, \ldots, X_m$.

For a user-chosen confidence level $1 - \beta \in (0, 1)$, we seek a data-dependent lower bound $\widetilde{J}_{\mathcal{M}}^\beta(\pi)$ computed from the $m$ observed statistics $X_1, \ldots, X_m$ such that

$$\Pr\left[ J_{\mathcal{M}}(\pi) \geq \widetilde{J}_{\mathcal{M}}^\beta(\pi) \right] \geq 1 - \beta, \tag{3}$$

where the probability is over the rollout randomness in $\mathcal{M}^\pi$. Our main result will then aggregate these *probabilistic* per-task certificates across tasks to obtain a guarantee of the form (2), explicitly accounting for the fact that each $\widetilde{J}_{\mathcal{M}}^\beta(\pi)$ is itself valid only with probability $1 - \beta$.

In the following, we discuss standard ways to construct such per-task bounds using one-sided confidence intervals and concentration inequalities (Boucheron et al., 2013). We focus on two common cases: (i) *binary* metrics, where $X_j \in \{0, 1\}$ (e.g., success/failure), and (ii) *bounded* real-valued metrics, where $X_j \in [a, b]$ for some $a \leq b$ (e.g., discounted return with bounded rewards).

**Binary metrics.** For binary metrics, $X_j \in \{0, 1\}$ and thus $X_j$ is Bernoulli with mean $p = \mathbb{E}[X_j] = J_{\mathcal{M}}(\pi)$. Let $s := \sum_{j=1}^{m} X_j$ be the number of successes. The *exact* one-sided lower confidence bound is the Clopper–Pearson bound (Clopper & Pearson, 1934):

$$\widetilde{J}_{\mathcal{M}}^\beta(\pi) := B\big(\beta; s, m - s + 1\big), \tag{4}$$

where $B(\beta; u, v)$ denotes the $\beta$-quantile of the $\mathrm{Beta}(u, v)$ distribution. Clopper–Pearson guarantees the nominal coverage $1 - \beta$ at any finite $m$, unlike normal-approximation intervals such as the Wald-interval (Wald, 1943), which can be under-covered in finite samples (Brown et al., 2001).

**Bounded real-valued metrics.** When $X_j \in [a, b]$, a range of concentration inequalities can be used to construct a one-sided lower confidence bound $\widetilde{J}_{\mathcal{M}}^\beta(\pi)$. A simple and widely used choice is Hoeffding's inequality (Hoeffding, 1963), which yields the following sound, distribution-free bound:

$$\widetilde{J}_{\mathcal{M}}^\beta(\pi) := \widehat{J}_{\mathcal{M}}(\pi) - (b - a)\sqrt{\frac{\ln(1/\beta)}{2m}}. \tag{5}$$

While easy to apply, Hoeffding's bound can be conservative in practice. We therefore also consider potentially tighter alternatives. The Dvoretzky–Kiefer–Wolfowitz (DKW) inequality (Dvoretzky et al., 1956; Massart, 1990) provides a uniform confidence band for the empirical CDF and can be converted into lower bounds on expectations. It is often less conservative when the samples concentrate away from the worst-case endpoints $a$ and $b$. We further consider empirical Bernstein bounds (Maurer & Pontil, 2009; Audibert

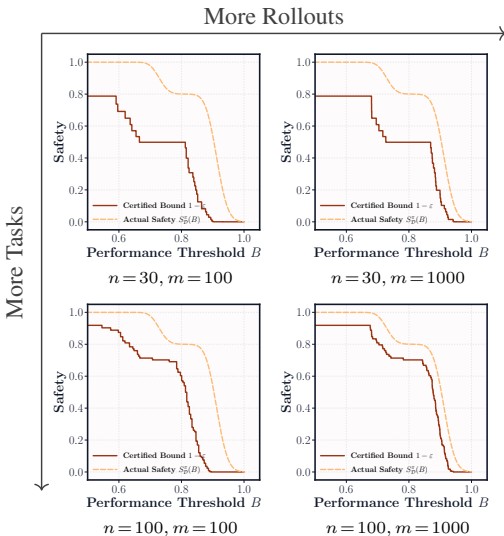

*Figure 3.* Illustration of the effect of the number of sampled tasks $n$ and rollouts per task $m$ on the certificate from Theorem 5.1.

et al., 2009), which adapt to the empirical variance and can substantially tighten bounds when rollout outcomes have low variance. Full details on the bounds are given in Appendix A.

### 5.2. Certifying Performance in Unseen Tasks

We now lift the per-task performance bounds from the previous section to a *distribution-level* guarantee for an unseen task $\mathcal{M} \sim \mathcal{D}$. Our main result provides a finite-sample certificate for the probability that $\pi$ meets a desired performance threshold on the unseen task, while accounting for *both* sources of uncertainty. The input is a set of i.i.d. sampled tasks $\mathbb{M} = \{\mathcal{M}_i \sim \mathcal{D}\}_{i=1}^n$ together with rollout-based per-task lower confidence bounds $\tilde{J}_{\mathcal{M}_i}^\beta(\pi)$ satisfying (3).

Crucially, unlike classical generalisation analyses, we do *not* observe the task performances $J_{\mathcal{M}_i}(\pi)$, but only *probabilistic* lower bounds on them. Our main result shows how to nevertheless obtain a *single* certificate for $S_{\mathcal{D}}^\pi(B)$ that is valid with user-chosen confidence $1 - \delta$ and *jointly* accounts for (i) finite task sampling from $\mathcal{D}$ and (ii) finite-rollout uncertainty within each sampled task $\mathcal{M}_i$.

The certified bound $1 - \varepsilon$ depends on how many of the per-task lower bounds fall below the performance level $B$. For a threshold $B \in \mathbb{R}$, we define

$$k(B) := |\{\tilde{J}_i \mid \tilde{J}_i < B\}|,$$

as the number of lower confidence bounds that fall below the performance threshold $B$. If $k(B) = n$ no sampled task attains a certified lower bound above $B$. In this case, the resulting certificate reduces to the trivial guarantee $S_{\mathcal{D}}^\pi(B) \geq 0$, or equivalently $\varepsilon = 1$.

**Theorem 5.1** (Safety Certificate). *Given a policy $\pi$, $n$ i.i.d. sample tasks $\mathbb{M} := \{\mathcal{M}_i \sim \mathcal{D}\}_{i=1}^n$, and per-task lower bounds $\tilde{J}_{\mathcal{M}_i}^\beta(\pi)$ such that*

$$\Pr\left[J_{\mathcal{M}_i}(\pi) \geq \tilde{J}_{\mathcal{M}_i}^\beta(\pi)\right] \geq 1 - \beta,$$

*for $\beta \in (0, 1)$. For any confidence $1 - \delta$, with $\delta \in (0, 1)$ and fixed performance threshold $B \in \mathbb{R}$, it holds that*

$$\Pr\left[S_{\mathcal{D}}^\pi(B) \geq 1 - \varepsilon\right] \geq 1 - \delta,$$

*where $\varepsilon \in [0, 1]$ is the unique solution to the equation*

$$\left(\sum_{i=K}^{n-k(B)} \binom{n - k(B)}{i}(1 - \beta)^i \beta^{n-k(B)-i}\right)$$
$$- \left(1 - \frac{\delta}{n + 1}\right) = \sum_{i=0}^{n-K} \binom{n}{i}\varepsilon^i(1 - \varepsilon)^{n-i}, \quad (6)$$

*and the bound is valid uniformly for any $0 \leq K \leq n - k(B)$.*

*Proof sketch.* The proof separates the uncertainty across tasks from the uncertainty within each task, and then combines them in a single finite-sample statement. Across tasks, we use an order-statistic generalisation argument: if we knew the true task performances $J_{\mathcal{M}_i}(\pi)$, the number $k(B)$ would control the probability that a fresh task drawn from $\mathcal{D}$ falls below the corresponding threshold. In our setting, however, we do not observe $J_{\mathcal{M}_i}(\pi)$; we only observe rollout-based lower bounds $\tilde{J}_{\mathcal{M}_i}^\beta(\pi)$ that are correct with probability at least $1 - \beta$. We therefore quantify the probability that sufficiently many of these per-task bounds are simultaneously valid: requiring at least $K$ of the $n - k(B)$ non-discarded bounds to hold yields the binomial term on the left-hand side of (6). Conditioning on this event, the order-statistic argument yields a bound on the violation probability $\varepsilon$, expressed by the binomial tail on the right-hand side of (6). By applying a union bound over possible values of $K$ we make the guarantee uniformly valid and allow for $K$ to be tuned for the tightest guarantee. Solving (6) for $\varepsilon$ gives a certificate that is valid under both task sampling and rollout estimation uncertainty. The full proof is given in Appendix C. $\qquad\square$

Theorem 5.1 includes two confidence parameters $\beta$ and $\delta$, reflecting the partitioning of the *confidence budget* across the two layers of uncertainty. The global confidence $1 - \delta$ represents the total reliability required by the user. This budget must cover both the risk of violated per-task lower confidence bounds, controlled by $\beta$, and the risk due to observing only $n$ tasks from $\mathcal{D}$, which is reflected in the resulting violation probability bound $\varepsilon$. While $\delta$ is typically fixed by safety requirements, $\beta$ acts as a tuning parameter: decreasing $\beta$ increases per-task confidence but makes

the individual bounds more conservative. Crucially, $\beta$ enters many standard per-task bounds only logarithmically (see Section 5.1), meaning high per-task confidence can be achieved with a modest loss in tightness.

A key feature of Theorem 5.1 is the auxiliary parameter $K$, which relaxes the need for all task-level bounds to hold simultaneously by requiring that only at least $K$ of the $n - k(B)$ relevant bounds are valid at once, correctly accounting for the potentially violated ones. The obtained guarantee is sound for any fixed choice of $\beta$, i.e., for any $\beta$ there exist choices of $K$ that yield a valid solution $\varepsilon$. If $\beta$ is too large relative to $n$ and $\delta$, then no non-trivial solution with $\varepsilon < 1$ exists, since the probability that at least $K$ per-task bounds are simultaneously valid drops below $1 - \delta$. In our experiments, choosing $\beta$ in the order of $\delta/n$ consistently yielded tight certificates. Since certification is computationally lightweight, our implementation searches over admissible $K$ to return the tightest bound. As shown in Section 6, this search is efficient and produces informative guarantees in practice. Appendix B details conditions for obtaining a non-trivial bound, discusses practical parameter selection, and analyses the effect of varying $\beta$ and $\delta$.

For fixed $\beta$, $\delta$, $n$, $B$, and $K$, Equation (6) can be solved efficiently for its unique solution $\varepsilon \in [0, 1]$. The entire left-hand side is a constant, while the right-hand side is a binomial tail probability. Hence $\varepsilon$ can be computed via bisection techniques, making the overall certification step lightweight.

Figure 3 illustrates how the certificate changes as we vary the number of rollouts per task and the number of tasks, for fixed confidence parameters $\delta$ and $\beta$. The certified safety bound $1 - \varepsilon$ is a step function of the threshold $B$, where each step corresponds to a change in $k(B)$, the number of per-task lower confidence bounds that fall below $B$.

Increasing the number of rollouts tightens the per-task lower confidence bounds, typically increasing their values. As a result, the step locations in $B$ move rightward. Equivalently, for any fixed threshold $B$, fewer per-task bounds fall below $B$, so the number $k(B)$ decreases and the certified safety improves. Increasing the number of tasks adds resolution: more tasks produce more order statistics and hence more (and smaller) steps in $B$. Moreover, more task samples sharpen the task-level generalisation step itself: for a fixed overall confidence $1 - \delta$, the uncertainty from sampling tasks decreases, which improves the resulting safety certificate. Finally, the flat region for small $B$ occurs because all per-task lower bounds exceed $B$, so $k(B)$ and thus the certificate remains unchanged over that range.

### 5.3. Next-Episode Guarantees

Our main results certify *task-level* performance: for a fresh task $\mathcal{M} \sim \mathcal{D}$, we lower bound the expected met-

ric $J_\mathcal{M}(\pi) = \mathbb{E}_{\tau \sim \mathcal{M}^\pi}[\mathcal{G}(\tau)]$. This notion is natural when a policy is deployed repeatedly within the same task, as it averages over rollouts' randomness, such as stochastic transitions, action choices or random initial conditions.

An alternative objective is to certify the outcome of a *single future episode*, which yields a guarantee under the joint draw of a new task and a single rollout in that task:

$$\Pr_{\substack{\mathcal{M} \sim \mathcal{D}, \\ \tau \sim \mathcal{M}^\pi}} \big(\mathcal{G}(\tau) \geq t\big) \geq 1 - \delta, \tag{7}$$

for a trajectory-level threshold $t$. Such *episodic* guarantees are most appropriate in *one-shot* settings where even a single failure is unacceptable, e.g., medical decision-making.

The discussed task-level guarantees address a different operational concern: *sustained* reliability within a new task. For example, an industrial robot deployed to a new warehouse may occasionally slip, yet what matters operationally is that its long-run throughput meets a performance bound. In these settings, insisting on worst-case single-episode behavior can be overly conservative: rare, fixable failures may be tolerable, while persistently low average performance is not.

The key technical distinction is that episodic- and task-level certification involve different uncertainty structures. Episodic certification treats the pair $(\mathcal{M}, \tau)$ as a single draw from the joint distribution induced by $\mathcal{M} \sim \mathcal{D}$ and $\tau \sim \mathcal{M}^\pi$. Consequently, there is only a single layer of uncertainty, and the relevant statistic is directly observable as $\mathcal{G}^*(\tau)$ in the finite-horizon case, or as a lower bound $\mathcal{G}^*(\tau[..h])$ computed from a finite prefix in the infinite-horizon case. Given i.i.d. samples of this statistic, a bound of the form (7) follows from standard concentration inequalities (see Section 5.1).

However, obtaining i.i.d. samples for (7) requires drawing tasks independently and evaluating the policy with a single rollout per task. Multiple rollouts from the same task are only conditionally independent and induce task-level dependence, violating the assumptions of standard concentration bounds and effectively overweighting the sampled tasks. As a result, episodic guarantees are inherently sample inefficient when the task distribution is unknown, as we can only leverage a single rollout per task.

For task-level certification, we bound $J_\mathcal{M}(\pi)$ for the next unseen task, which is a latent expectation that is never observed directly. This introduces a second layer of uncertainty: we must control both (i) within-task estimation error from finitely many rollouts and (ii) across-task sampling error from observing only finitely many tasks. Theorem 5.1 jointly addresses both layers and the final certificate is sound even under heterogeneous data collection, and it exploits all $m_i$ available rollouts per task rather than only a single one.

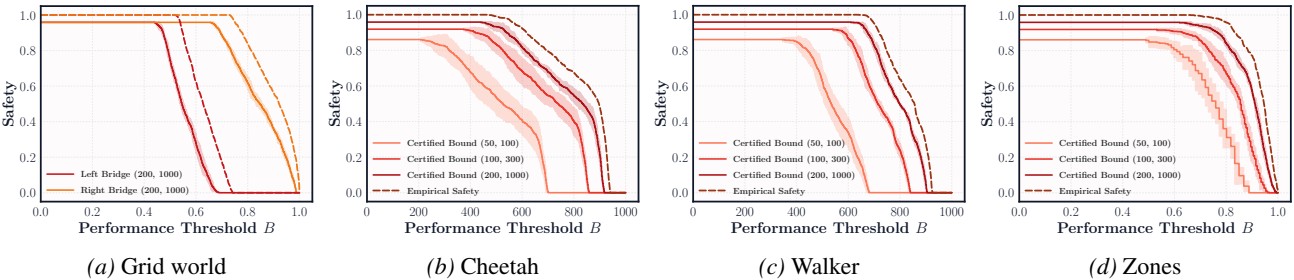

*Figure 4.* Empirical safety (dashed) and certified safety (solid) for trained policies. In *BridgeWorld* we compare fixed left- vs. right-bridge policies. In *Cheetah*, *Walker*, and *Zones* we vary the number of sampled tasks $n$ and rollouts per task $m$, indicated by $(n, m)$.

## 6. Experiments

We conduct an experimental investigation of our certification guarantees across a range of MTRL algorithms and environments.[1] Our experiments address three questions: **(1)** How tight are the certificates, i.e., how do bounds computed from limited data compare to empirical safety estimated from substantially larger evaluation sets? **(2)** Do the certificates remain informative in high-dimensional continuous-control domains and across diverse task distributions? **(3)** How do the certified safety levels vary with the number of sampled tasks and the number of rollouts collected per task?

### 6.1. Experimental Setup

**Environments.** We evaluate our method across environments of varying complexity. We begin with a grid-world navigation task inspired by Greenberg et al. (2023). The agent must reach a target by crossing one of two bridges. Both bridges are slippery, with different slip probabilities drawn from task-specific distributions. As a result, policies that take the longer route via the right bridge are typically more reliable. We additionally consider standard high-dimensional continuous-control benchmarks, *Cheetah* and *Walker* (Tunyasuvunakool et al., 2020), where the policy must learn locomotion gaits. Tasks are generated by varying physical parameters of the agent, specifically its mass and size. Finally, we study multi-task policies trained with *DeepLTL* (Jackermeier & Abate, 2025) in the *Zones* environment (Vaezipoor et al., 2021). There, the agent navigates between different zones observed via a lidar sensor, and tasks are specified as linear temporal logic (LTL) formulae (Pnueli, 1977) describing which zones to reach and avoid, and in what order. Appendix D.1 provides visualisations and further details on the environments and task distributions.

**Multi-task policies.** We first obtain multi-task policies to which we then apply our certification procedure. In the grid-world experiment, we consider two simple tabular policies that deterministically choose the left or right bridge,

respectively. In the continuous-control domains, we train task-conditioned policies using proximal policy optimisation (PPO; Schulman et al., 2017).

In *Cheetah* and *Walker*, we sample mass and body-scaling parameters $\mu \in \mathbb{R}^2$ at the start of each episode and condition the policy by concatenating the observation $o \in \mathbb{R}^n$ with $\mu$. In *Zones*, we train an LTL-conditioned multi-task policy using DeepLTL (Jackermeier & Abate, 2025), which maps LTL specifications to sequential task representations processed by a recurrent neural network (RNN). The RNN and policy are trained jointly via goal-conditioned PPO. Further training details are provided in Appendix D.2.

**Experimental protocol.** We compute per-task lower confidence bounds $\tilde{J}^{\beta}_{\mathcal{M}_i}(\pi)$ from $m$ rollouts in each sampled task $\mathcal{M}_i$. In grid-world we use $n = 200$ tasks with $m = 1000$ rollouts per task. In *Zones*, *Cheetah*, and *Walker* we vary $n$ and $m_i$ to study the effect of data on the certificates. For grid-world and *Zones* we certify binary success using the one-sided Clopper–Pearson bound (Clopper & Pearson, 1934); for *Cheetah* and *Walker* we certify undiscounted return in $[0, 1000]$ using the one-sided empirical Bernstein bound (Maurer & Pontil, 2009). We fix $\beta = 1 \times 10^{-4}$ and $\delta = 0.01$ throughout. For each $(n, m)$ setting, we repeat task and rollout sampling over 10 random seeds and report the mean certificate together with a $\pm 1$ standard deviation band. Since we average the certificates across runs, the resulting curves need not appear as exact step functions. To assess soundness and tightness, we estimate *empirical safety* over a grid of thresholds from a much larger evaluation set. This evaluation data is used only as a near-ground-truth reference and is not required for certification.

### 6.2. Results

Figure 4 reports safety certificates produced by our method for the trained multi-task policies across all evaluation environments, together with empirical safety curves estimated from a large-scale dataset of $10^7$ trajectories per environment. The certified bounds are sound in all cases, matching the theoretical guarantees. Moreover, they closely track the

---

[1]Code available at: github.com/mathiasj33/mtrl-guarantees

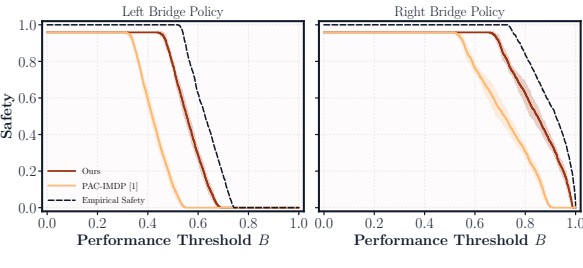

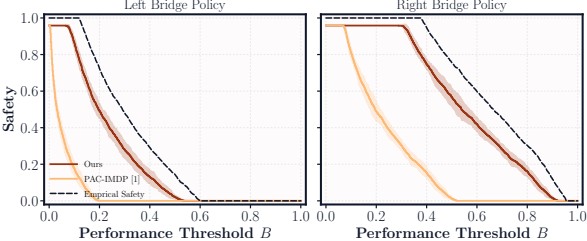

*Figure 5.* Comparison of PAC-IMDP-based guarantees [1] and our guarantees across two discrete, finite BridgeWorld domains. The upper panel reproduces the benchmark from the paper, while the lower panel shows the same comparison on a larger grid.

empirical safety despite being computed from substantially fewer rollouts, and are particularly tight in the high-safety regime that is most relevant for certification.

In the grid-world example (Figure 4a), the certificates correctly reflect that the more conservative policy, which takes the right bridge with typically lower slip probabilities, achieves higher safety. In the high-dimensional continuous-control domains (Figures 4b to 4d), the method continues to produce informative and tight guarantees from realistic evaluation budgets. This holds both when tasks vary through environment parameters (as in *Cheetah* and *Walker*) and under more heterogeneous task families, such as task distributions induced by LTL specifications in *Zones*.

**Further results.** Additional results are provided in Appendix D. We study the effect of varying the confidence parameters $\beta$ and $\delta$, compare different choices of per-task lower confidence bounds, and report empirical performance distributions alongside the certified curves. We also include rollout videos of the trained policies.

### 6.3. Comparison to PAC-IMDP Guarantees

We further conduct an experimental comparison of our method to the approach of Schnitzer et al. (2025). They explicitly construct *interval MDPs* (IMDPs; Iyengar, 2005; Nilim & Ghaoui, 2005) that over-approximate the environmental uncertainty, and derive performance guarantees from these uncertainty estimates. As their approach approximates the underlying model rather than the performance itself, it

does not require the monotonicity assumption on trajectory metrics. However, it is restricted to finite-state systems with a known transition structure, and the tightness of the guarantees deteriorates with growing model complexity since the confidence budget has to be distributed over an increasingly large number of transitions. By contrast, we require no assumptions about MDP structure and derive guarantees directly from observed rollouts, making our approach applicable to complex and continuous-state systems.

Figure 5 compares the guarantees computed by our approach and by the PAC-IMDP approximation of Schnitzer et al. (2025) on the discrete grid-world navigation task. The upper panel uses the BridgeWorld instance from our main grid-world experiment, while the lower panel repeats the same comparison on a larger version of the domain with substantially more states and transitions. We observe that our bounds are consistently tighter for the same number of sampled tasks and rollouts. Moreover, our method scales more favourably with respect to domain complexity: whereas the PAC-IMDP bounds become substantially more conservative in the larger domain, our certificates remain similarly close to the empirical safety curve as in the smaller domain.

## 7. Conclusion

We have presented a certification method for multi-task reinforcement learning that yields high-confidence safety guarantees for a fixed policy on a new task drawn from an unknown task distribution. Our result composes rollout-based per-task lower confidence bounds with task-level generalisation, producing an interpretable certificate for user-chosen performance thresholds. Experiments show that the certificates are sound and remain informative in high-dimensional continuous-control domains.

Future work includes Sim-to-Real transfer, where the deployment task may not be drawn from the same distribution as the tasks used to derive the certificate, incorporating a quantified notion of distribution shift into the guarantee.

## Acknowledgements

This work was supported in part by the UKRI Erlangen AI Hub on Mathematical and Computational Foundations of AI (No. EP/Y028872/1). MJ is funded by the EPSRC Centre for Doctoral Training in Autonomous Intelligent Machines and Systems (EP/S024050/1).

## Impact Statement

This paper presents work whose goal is to advance the field of Machine Learning. There are many potential societal consequences of our work, none which we feel must be specifically highlighted here.

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

# A. Concentration Inequalities for Per-Task Lower Confidence Bounds

We provide additional details on the per-task lower confidence bounds used in Section 5.1. Throughout, we consider i.i.d. rollout statistics $X_1, \ldots, X_m$ obtained by executing $\pi$ in a fixed task $\mathcal{M}$, and we write $\mu := \mathbb{E}[X_1]$. Our goal is to construct a data-dependent lower bound $\tilde{\mu}^\beta$ such that

$$\Pr\big[\mu \geq \tilde{\mu}^\beta\big] \geq 1 - \beta, \tag{8}$$

for a user-chosen $\beta \in (0, 1)$. In the finite-horizon case, $\mu = J_{\mathcal{M}}(\pi)$. In the infinite-horizon case, by monotonicity we have $\mu \leq J_{\mathcal{M}}(\pi)$, so any lower bound on $\mu$ remains a valid lower bound on $J_{\mathcal{M}}(\pi)$.

## A.1. Binary metrics: Clopper–Pearson

Assume $X_j \in \{0, 1\}$, so $X_j \sim \text{Bernoulli}(p)$ with mean $\mu = p$. Let $s := \sum_{j=1}^m X_j$ be the number of successes. The Clopper–Pearson interval is obtained by inverting the exact binomial test and provides guaranteed coverage for any finite sample size without relying on asymptotic approximations (Clopper & Pearson, 1934; Brown et al., 2001). The one-sided $(1 - \beta)$ lower confidence bound is

$$\tilde{\mu}^\beta \;=\; B(\beta; \, s, \, m - s + 1), \tag{9}$$

where $B(\beta; u, v)$ denotes the $\beta$-quantile of the $\text{Beta}(u, v)$ distribution. Concretely, $\text{Beta}(u, v)$ is the distribution on $[0, 1]$ with density proportional to $x^{u-1}(1 - x)^{v-1}$, and its quantile function can be computed using standard numerical routines (Kotz et al., 2000). Under i.i.d. Bernoulli sampling, the bound in (9) satisfies (8) for all $m$ and all $p \in [0, 1]$. In practice, Clopper–Pearson is a safe default for binary outcomes, unlike normal-approximation intervals based on the central limit theorem, it does not under-cover (Brown et al., 2001; Wald, 1943).

## A.2. Bounded metrics: Hoeffding

Assume $X_j \in [a, b]$ almost surely, with unknown distribution and mean $\mu$. Let $\hat{\mu} := \frac{1}{m}\sum_{j=1}^m X_j$. Hoeffding's inequality implies that, for any $\epsilon > 0$,

$$\Pr[\mu \leq \hat{\mu} - \epsilon] \leq \exp\left(-\frac{2m\epsilon^2}{(b - a)^2}\right)$$

for i.i.d. samples (Hoeffding, 1963; Boucheron et al., 2013). Setting the right-hand side to $\beta$ and solving for $\epsilon$ yields the one-sided lower bound

$$\tilde{\mu}^\beta \;=\; \hat{\mu} \;-\; (b - a)\sqrt{\frac{\ln(1/\beta)}{2m}}. \tag{10}$$

Hoeffding is fully distribution-free and depends only on the known range $[a, b]$, making it broadly applicable. Its downside is that it is calibrated to a worst-case distribution over $[a, b]$ and can therefore be conservative when the rollout outcomes concentrate well away from the endpoints (Boucheron et al., 2013).

## A.3. Bounded metrics: Empirical Bernstein

Empirical Bernstein bounds tighten Hoeffding-type guarantees by adapting to the empirical variance. Assume $X_j \in [a, b]$ almost surely and define the unbiased sample variance

$$\hat{V} \;:=\; \frac{1}{m - 1}\sum_{j=1}^m (X_j - \hat{\mu})^2.$$

A one-sided empirical Bernstein inequality (e.g. Maurer & Pontil (2009); see also Audibert et al. (2009)) implies that, with probability at least $1 - \beta$,

$$\mu \;\geq\; \hat{\mu} \;-\; \sqrt{\frac{2\hat{V}\ln(2/\beta)}{m}} \;-\; \frac{7(b - a)\ln(2/\beta)}{3(m - 1)}. \tag{11}$$

We use the right-hand side of (11) as $\tilde{\mu}^\beta$. Compared to Hoeffding, the leading term scales with the empirical standard deviation $\sqrt{\hat{V}}$ rather than the full range $(b - a)$. This can yield significantly tighter bounds when rollout outcomes have low variance, i.e., when the policy is reliable and returns concentrate, while retaining finite-sample coverage under the same i.i.d. and boundedness assumptions.

## A.4. Bounded metrics: DKW-based lower bounds

The Dvoretzky–Kiefer–Wolfowitz (DKW) inequality provides a uniform confidence band for the empirical CDF. Let $F$ be the CDF of $X_1$ and $\hat{F}$ the empirical CDF of $X_1, \dots, X_m$, i.e.,

$$\hat{F}(x) \;:=\; \frac{1}{m} \sum_{j=1}^{m} \not\Vdash\{X_j \leq x\}.$$

DKW states that for any $\epsilon > 0$,

$$\Pr\left[\sup_{x \in \mathbb{R}} |\hat{F}(x) - F(x)| > \epsilon\right] \leq 2e^{-2m\epsilon^2}$$

for i.i.d. samples (Dvoretzky et al., 1956; Massart, 1990). Setting $\epsilon := \sqrt{\frac{\ln(2/\beta)}{2m}}$ yields, with probability at least $1 - \beta$,

$$F(x) \;\leq\; \hat{F}(x) + \epsilon, \text{ for all } x. \tag{12}$$

To turn this band into a lower confidence bound on the mean for bounded $X \in [a, b]$, we use the identity

$$\mu \;=\; \int_a^b \big(1 - F(x)\big)\, dx,$$

and pessimistically upper bound $F$ by $\hat{F} + \epsilon$ in the integrand. This yields the sound lower bound

$$\mu \;\geq\; \int_a^b \big(1 - \hat{F}(x) - \epsilon\big)_+ \, dx,$$

which can be computed from the sorted samples $X_{(1)} \leq \cdots \leq X_{(m)}$.

A discrete form follows from interpreting (12) as a uniform control of the CDF: with probability at least $1 - \beta$, the true distribution can place at most an additional $\epsilon$ probability mass below any threshold compared to the empirical CDF (Budde et al., 2025). A worst-case lower bound on the mean is therefore obtained by shifting an $\epsilon$-fraction of mass to the smallest possible value $a$. Approximating this $\epsilon$-fraction by $\ell := \lceil m\epsilon \rceil$ samples yields the conservative bound obtained by replacing the largest $\ell$ samples by $a$ and averaging. Concretely, let

$$q_m \;:=\; \sqrt{\frac{\ln(2/\beta)}{2m}}, \quad \ell \;:=\; \lceil mq_m \rceil,$$

and define

$$\tilde{\mu}^\beta \;:=\; \frac{1}{m}\left(\sum_{j=1}^{m-\ell} X_{(j)} \;+\; \ell \cdot a\right). \tag{13}$$

In contrast to Hoeffding, which depends only on the range $[a, b]$, DKW leverages the empirical distribution shape through the order statistics. It can be noticeably tighter when the samples concentrate away from the worst-case endpoints, since it effectively certifies that only a small fraction of probability mass could lie below the observed values while remaining consistent with the DKW band (Boucheron et al., 2013).

## A.5. Finite-sample soundness

All bounds above are non-asymptotic and satisfy the coverage requirement (8) under their stated assumptions, i.e., i.i.d. rollouts within a fixed task, and either Bernoulli or bounded outcomes. In particular, they do not rely on central-limit approximations and remain valid at finite sample sizes. This is important for certification: normal-approximation intervals such as the Wald interval can under-cover in finite samples (Wald, 1943; Brown et al., 2001; Budde et al., 2025) and may therefore yield unsound guarantees when used as building blocks in distribution-level certificates.

# B. Relation of Confidence Parameters and Parameter Selection

This appendix elaborates on the roles of the confidence parameters $\beta$ and $\delta$ in Theorem 5.1, the auxiliary parameter $K$, and the feasibility conditions under which the theorem yields a non-vacuous certificate. We also summarise practical parameter-selection strategies and the post-hoc search used in our implementation.

## B.1. Two-layer confidence budget: within-task and across-task uncertainty

Theorem 5.1 combines two statistically distinct sources of uncertainty. First, for each sampled task $\mathcal{M}_i \sim \mathcal{D}$, we do not observe the true performance $J_{\mathcal{M}_i}(\pi)$, but only a rollout-based lower confidence bound $\tilde{J}^\beta_{\mathcal{M}_i}(\pi)$ that satisfies

$$\Pr\left[J_{\mathcal{M}_i}(\pi) \geq \tilde{J}^\beta_{\mathcal{M}_i}(\pi)\right] \geq 1 - \beta.$$

The parameter $\beta$ therefore controls the *within-task* probability that the bound fails to lower-bound the true performance. Second, we ultimately seek a *distribution-level* statement about an unseen task $\mathcal{M} \sim \mathcal{D}$, based on only $n$ sampled tasks; this is the *across-task* generalisation component, and its uncertainty is reflected in the resulting violation probability bound $\varepsilon$.

The global confidence parameter $\delta$ controls the probability that the overall certificate fails:

$$\Pr\left[S^\pi_\mathcal{D}(B) \geq 1 - \varepsilon\right] \geq 1 - \delta.$$

Intuitively, the confidence budget $\delta$ must cover both (i) the possibility that some task-level lower bounds are invalid and (ii) the error induced by having only $n$ tasks to represent $\mathcal{D}$. Theorem 5.1 makes this interaction explicit through a feasibility condition and through the auxiliary parameter $K$, described next.

## B.2. The role of $K$: relaxing simultaneous validity

A key feature of Theorem 5.1 is that it does *not* require all task-level lower bounds to hold simultaneously. Instead, it introduces an auxiliary integer $K \in [0, n - k(B)]$ specifying how many of the $n - k(B)$ relevant per-task bounds must be correct *at once* in order to lift them to a distribution-level certificate. Larger $K$ imposes a stricter simultaneous-validity requirement, while smaller $K$ tolerates occasional failures of per-task confidence bounds. This relaxation matters whenever $n$ is large or $\beta$ is not extremely small: even if each per-task bound fails with probability at most $\beta$, the probability that *all* bounds are simultaneously correct can be much smaller than $1 - \delta$.

Formally, the left-hand side of (6) is the binomial tail probability

$$\sum_{i=K}^{n-k(B)} \binom{n - k(B)}{i} (1 - \beta)^i \beta^{n-k(B)-i} \;=\; \Pr\left[\mathrm{Bin}(n - k(B), 1 - \beta) \geq K\right], \tag{14}$$

i.e., the probability that at least $K$ of the $n - k(B)$ relevant per-task bounds are valid, under independent per-task validity events. The parameter $K$ therefore provides a principled trade-off between strict simultaneous validity and robustness to a limited number of per-task bound failures.

As a reference point, the conservative "all-at-once" requirement corresponds to setting $K = n - k(B)$, i.e., demanding that all relevant per-task bounds hold simultaneously. This yields

$$\Pr\left[\mathrm{Bin}(n - k(B), 1 - \beta) \geq n - k(B)\right] = (1 - \beta)^{n-k(B)},$$

so the corresponding condition reduces to $(1 - \beta)^{n-k(B)} \geq 1 - \delta$. Theorem 5.1 generalises this by allowing $K < n - k(B)$, which permits a small number of per-task bound failures while preserving a sound global guarantee. In practice, we can often take $K$ close to $n - k(B)$: since the binomial tail probability in (14) approaches 1 rapidly as $K$ drops slightly below its maximum, the relaxation typically incurs only a small penalty while enabling tight certificates.

## B.3. Feasibility and existence of a non-vacuous certificate

Equation (6) defines $\varepsilon \in [0, 1]$ implicitly by equating a constant (the left-hand side) to a binomial tail probability in $\varepsilon$ (the right-hand side). Since the right-hand side always lies in $[0, 1]$, a necessary condition for Eq. (6) to admit any solution $\varepsilon \in [0, 1]$ is that the left-hand side is nonnegative. This yields the feasibility constraint

$$\sum_{i=K}^{n-k(B)} \binom{n - k(B)}{i} (1 - \beta)^i \beta^{n-k(B)-i} \;\geq\; 1 - \delta. \tag{15}$$

In words, the probability that at least $K$ of the $n - k(B)$ relevant task-level confidence bounds are simultaneously valid must exceed the global confidence requirement $1 - \delta$. When (15) fails, Equation (6) cannot yield a nontrivial solution and the certificate collapses to the vacuous guarantee $\varepsilon = 1$.

Condition (15) also clarifies the interaction between $\beta$ and $\delta$. For fixed $n$ and $B$, increasing $\beta$ decreases the probability that many task-level bounds are simultaneously valid, and therefore tightens the feasible range of $K$, possibly to values that yield only vacuous certificates. Conversely, decreasing $\beta$ makes feasibility easier to satisfy, allowing larger $K$, but increases conservatism of the per-task bounds and can thereby increase $k(B)$.

In many standard concentration-based lower confidence bounds, the confidence parameter $\beta$ affects the width of the bound only logarithmically. For example, for bounded random variables, Hoeffding-style bounds yield deviations scaling as $\sqrt{\ln(1/\beta)}$. As a consequence, reducing $\beta$ changes $\tilde{J}^{\beta}_{\mathcal{M}_i}(\pi)$ only moderately. This empirical robustness motivates choosing $\beta$ conservatively to ensure that within-task estimation failures do not dominate the overall confidence budget.

### B.4. Parameter selection and post-hoc optimisation

In typical safety-critical use, the practitioner first fixes $\delta$ to reflect the desired reliability of the final certificate. The remaining parameters play different roles: $\beta$ determines the confidence level of the per-task bounds, while $K$ is an internal tuning parameter of Theorem 5.1.

**Choosing $\beta$.** We treat $\beta$ as a user-specified (or pre-specified) confidence level for the per-task certificates. A simple conservative guideline is to scale it inversely with the number of sampled tasks, e.g., $\beta \lesssim \delta/n$. This keeps the probability of accumulating many per-task failures small relative to the overall failure budget, while typically only mildly loosening the per-task bounds due to their logarithmic dependence on $1/\beta$. Since $\beta$ controls the nominal coverage of the per-task bounds, we fix it independently of the observed rollouts.

**Optimising over $K$.** In contrast, $K$ can be selected post hoc without compromising validity, since Theorem 5.1 is sound jointly for *any* admissible $K \in \{0, 1, \ldots, n - k(B)\}$. Accordingly, for a fixed $\beta$ we evaluate the certificate over all admissible $K$ and return the smallest feasible $\varepsilon$, i.e., among those satisfying (15). This selection is safe because it amounts to choosing the tightest instantiation among a family of valid certificates provided by the theorem.

In practice, the conservative guideline for $\beta$ works well, and the search over $K$ is inexpensive. Section 6 and the extended experiments in Appendix D confirm that this procedure reliably yields tight, non-vacuous certificates.

## C. Proof of Theorem 5.1

We prove Theorem 5.1 by decoupling two sources of uncertainty: (i) *rollout-level* uncertainty in the per-task certificates, and (ii) *task-level* generalisation across i.i.d. tasks. We then combine these layers into a single finite-sample guarantee. Finally, we make the statement *uniform in $K$* (thus permitting post-hoc, data-dependent selection of the number of retained tasks) by a union bound with confidence allocation $\delta/(n + 1)$.

Our proof follows the scenario-based generalisation analysis of Schnitzer et al. (2025). The main adaptation required here is that our per-task constraints are not given by model-based overapproximations, but are instead constructed from *finite rollouts* via lower confidence bounds. Accordingly, we first lift the rollout-level uncertainty into probabilistic task-wise constraints, and then apply the scenario approach across tasks. A key technical point beyond Schnitzer et al. (2025) is that we preserve finite-sample validity under a data-dependent choice of $K$ by explicitly uniformising over $K$.

We briefly recall the scenario approach (Campi & Garatti, 2018). Fix (i) a linear objective $c^{\top}x$, (ii) an admissible set $C \subseteq \mathbb{R}^d$, (iii) a family of convex constraint sets $\{C_\theta \subseteq \mathbb{R}^d \mid \theta \in \Theta\}$ indexed by an uncertain parameter $\theta$, and (iv) a probability measure $\mathbb{P}$ over $\Theta$. Given i.i.d. samples $\theta_1, \ldots, \theta_N \sim \mathbb{P}$, the associated scenario program is

$$\max_{x \in C} \quad c^{\top}x$$
$$\text{subject to} \quad x \in \bigcap_{i=1}^{N} C_{\theta_i}. \tag{16}$$

In our instantiation we take $d = 1$ and $c^{\top}x = x$. Let $x^*$ denote the optimal solution of (16). The scenario approach bounds the *violation probability*

$$V(x) = \mathbb{P}\{\theta \in \Theta : x \notin C_\theta\}, \tag{17}$$

i.e., the probability that the solution violates a fresh, unseen constraint. In our setting, $V(x)$ corresponds to the risk $\varepsilon$ that a policy fails to satisfy a target performance guarantee $B$ on a new task $\mathcal{M} \sim \mathcal{D}$.

The analysis in Schnitzer et al. (2025) provides bounds on $V(x^*)$ even when the constraints $C_{\theta_i}$ are only available through *probabilistic relaxations* of the true (unobservable) constraints. In our case, these probabilistic constraints arise from per-task rollout-based lower confidence bounds $\tilde{J}^\beta_{\mathcal{M}_i}(\pi)$ for the true performances $J_{\mathcal{M}_i}(\pi)$.

In our setting, $\theta$ is a task $\mathcal{M} \sim \mathcal{D}$ and we seek a lower performance guarantee $B$ for a fixed policy $\pi$ on unseen tasks. For each task $\mathcal{M}$, define the (unobservable) true constraint

$$C_{\mathcal{M}} := (-\infty, J_{\mathcal{M}}(\pi)], \tag{18}$$

so that $B \in C_{\mathcal{M}}$ is equivalent to $J_{\mathcal{M}}(\pi) \geq B$. Given rollouts on $\mathcal{M}$, we compute a one-sided lower confidence bound $\tilde{J}^\beta_{\mathcal{M}}(\pi)$ satisfying

$$\mathbb{P}\left\{\tilde{J}^\beta_{\mathcal{M}}(\pi) \leq J_{\mathcal{M}}(\pi)\right\} \geq 1 - \beta, \tag{19}$$

where the probability is over the rollout randomness (conditionally on $\mathcal{M}$). This induces an *observable* constraint set

$$\hat{C}_{\mathcal{M}} := (-\infty, \tilde{J}^\beta_{\mathcal{M}}(\pi)]. \tag{20}$$

By (19), each $\hat{C}_{\mathcal{M}}$ is an inner approximation of $C_{\mathcal{M}}$ with probability at least $1 - \beta$:

$$\mathbb{P}\{\hat{C}_{\mathcal{M}} \subseteq C_{\mathcal{M}}\} \geq 1 - \beta. \tag{21}$$

We now restate the key scenario result from Schnitzer et al. (2025) in a form aligned with our notation; we will subsequently instantiate it with the mapping above and then add uniformity over $K$.

**Theorem C.1** (Scenario bound with probabilistic inner constraints. Restated from Schnitzer et al. (2025))**.** *Fix a policy* $\pi$ *and let* $\mathcal{M}_1, \ldots, \mathcal{M}_n \sim \mathcal{D}$. *For each* $i$, *assume we can compute an* observable *inner approximation* $\hat{C}_{\mathcal{M}_i} \subseteq \mathbb{R}$ *of the (unobservable) true constraint set* $C_{\mathcal{M}_i}$ *such that*

$$\Pr[\hat{C}_{\mathcal{M}_i} \subseteq C_{\mathcal{M}_i}] \geq 1 - \beta,$$

*where the probability is over the randomness used to construct* $\hat{C}_{\mathcal{M}_i}$ *(e.g. rollout noise), conditional on* $\mathcal{M}_i$. *Assume all constraint sets are intervals of the form* $(-\infty, b]$. *Let* $\hat{x}^\star$ *be the solution returned by the scenario program using the observable constraints* $\hat{C}_{\mathcal{M}_1}, \ldots, \hat{C}_{\mathcal{M}_n}$, *and let* $r(\pi, \hat{x}^\star)$ *denote its violation probability on a fresh task, i.e.*

$$r(\pi, \hat{x}^\star) := \Pr_{\mathcal{M} \sim \mathcal{D}}[\hat{x}^\star \notin C_{\mathcal{M}}].$$

*Then for any confidence level* $1 - \eta$ *(with* $\eta \in (0, 1)$*) and any integer* $K \in \{0, \ldots, n\}$,

$$\Pr[r(\pi, \hat{x}^\star) \leq \varepsilon] \geq 1 - \eta, \tag{22}$$

*where* $\varepsilon \in [0, 1]$ *is the unique solution to*

$$\left(\sum_{i=K}^{n} \binom{n}{i}(1-\beta)^i \beta^{n-i}\right) - (1-\eta) = \sum_{i=0}^{n-K} \binom{n}{i}\varepsilon^i(1-\varepsilon)^{n-i}. \tag{23}$$

$\square$

We now instantiate Theorem C.1 in our setting. Since we certify a *fixed* performance threshold $B$ (rather than solving an optimisation problem), the relevant "decision variable" is the scalar $x = B$ itself. Recall that for each task $\mathcal{M}$ we defined the true and observable constraint sets

$$C_{\mathcal{M}} := (-\infty, J_{\mathcal{M}}(\pi)] \quad \text{and} \quad \hat{C}_{\mathcal{M}} := (-\infty, \tilde{J}^\beta_{\mathcal{M}}(\pi)],$$

so that $B \in C_{\mathcal{M}}$ is equivalent to $J_{\mathcal{M}}(\pi) \geq B$, and by (19) each $\hat{C}_{\mathcal{M}}$ is an inner approximation of $C_{\mathcal{M}}$ with probability at least $1 - \beta$. Let $\tilde{J}_i := \tilde{J}^\beta_{\mathcal{M}_i}(\pi)$ and recall

$$k(B) := \left|\{\tilde{J}_i \mid \tilde{J}_i < B\}\right|.$$

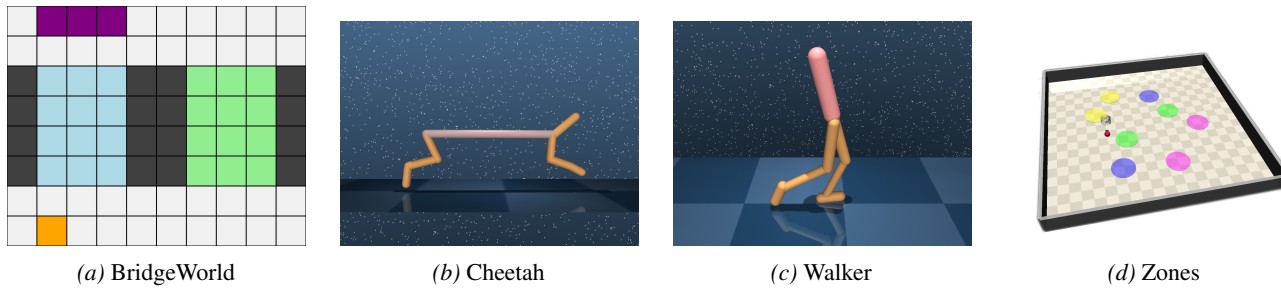

*(a)* BridgeWorld          *(b)* Cheetah          *(c)* Walker          *(d)* Zones

*Figure 6.* Environment visualisations.

If $k(B) = n$, then no sampled task attains a certified lower bound above $B$ and the certificate reduces to the trivial guarantee $S^\pi_{\mathcal{D}}(B) \geq 0$. Hence assume $k(B) \leq n - 1$. We apply Theorem C.1 with the *discard count* set to $\ell := k(B)$, i.e., we discard exactly those sampled tasks whose observed inner constraints cannot contain $B$ (since $\tilde{J}_i < B$), and retain the remaining $n - k(B)$ tasks for which $B \in \hat{C}_{\mathcal{M}_i}$ holds by construction. Among the retained tasks, the event $\hat{C}_{\mathcal{M}_i} \subseteq C_{\mathcal{M}_i}$ implies $B \in C_{\mathcal{M}_i}$, and these inner-approximation events hold independently with probability at least $1 - \beta$. Letting $\varepsilon := \Pr_{\mathcal{M} \sim \mathcal{D}}[J_{\mathcal{M}}(\pi) < B] = 1 - S^\pi_{\mathcal{D}}(B)$, Theorem C.1 therefore yields the following: for any fixed $K \in \{0, \ldots, n - k(B)\}$ and any confidence level $1 - \eta$,

$$\Pr\left[S^\pi_{\mathcal{D}}(B) \geq 1 - \varepsilon\right] \geq 1 - \eta,$$

where $\varepsilon \in [0, 1]$ is the unique solution of the equation obtained from (23) by replacing $n$ with $n - k(B)$ in the left-hand binomial tail (corresponding to the $n - k(B)$ retained tasks), namely

$$\left(\sum_{i=K}^{n-k(B)} \binom{n - k(B)}{i}(1 - \beta)^i \beta^{n-k(B)-i}\right) - (1 - \eta) = \sum_{i=0}^{n-K} \binom{n}{i}\varepsilon^i(1 - \varepsilon)^{n-i}. \qquad (24)$$

Finally, to permit *post-hoc* (data-dependent) selection of $K$, we uniformise the statement over all $K \in \{0, \ldots, n\}$ by a union bound. Allocating confidence $\eta := \delta/(n + 1)$ to each of the at most $n + 1$ choices of $K$ implies that, with probability at least $1 - \delta$, the bound $S^\pi_{\mathcal{D}}(B) \geq 1 - \varepsilon$ holds *simultaneously* for every admissible $K \in \{0, \ldots, n - k(B)\}$, where $\varepsilon$ is defined as the (unique) solution of (6) with $\eta = \delta/(n + 1)$. This proves Theorem 5.1. $\qquad \square$

## D. Experimental Details

### D.1. Environments

**BridgeWorld (grid world).**   Figure 6a shows the grid navigation domain. The agent must reach a goal by crossing either a left or a right bridge. When the agent is on bridge $b \in \{L, R\}$, it slips with probability $p_b$, executing a left move instead of the chosen action. We sample bridge slipperiness as $p_L \sim \mathcal{U}(0.2, 0.3)$ and $p_R \sim \mathcal{U}(0, 0.2)$.

**Continuous control (Cheetah and Walker).**   We use multi-task variants of *Cheetah* and *Walker* from the DeepMind Control Suite (Tunyasuvunakool et al., 2020), simulated with the MuJoCo physics engine (Todorov et al., 2012). Rewards are bounded by 1 per step; with episode length 1000, the undiscounted return lies in $[0, 1000]$. Tasks correspond to morphology parameters $\tau \in \mathbb{R}^2$ that scale mass and body size; we sample $\tau$ and include it in the observation by concatenation. Following RoML (Greenberg et al., 2023), we draw $\log(\tau)$ component-wise from $\mathcal{U}(-1, 1)$.

**Zones.**   We evaluate on *ZoneEnv* (Vaezipoor et al., 2021), a high-dimensional robotic navigation domain for LTL-guided multi-task RL (Figure 6d). The agent observes coloured zones via a lidar sensor in addition to proprioceptive state, and acts in a continuous action space. Zones are randomly placed at the beginning of each episode. Tasks are given as LTL formulae (Pnueli, 1977) over zone propositions, specifying which zones to reach and avoid over time.

We consider a reach–avoid task family with formula depth between 1 and 3. At each stage we uniformly sample 1–2 target zones to reach and 0–3 zones to avoid, composing these into a sequential specification. An example task is $\neg(\text{green} \vee \text{yellow})\, \mathsf{U}\, (\text{purple} \wedge (\neg \text{blue}\, \mathsf{U}\, \text{green}))$.

*Table 1.* PPO parameters for continuous control benchmarks.

| Parameter | Value |
|---|---|
| Total Timesteps | 60,000,000 |
| Number of Environments | 2,048 |
| Batch Size | 1,024 |
| Steps per Update | 30 |
| Minibatches | 32 |
| Epochs per Batch | 16 |
| Learning Rate | 0.001 |
| Discount Factor ($\gamma$) | 0.995 |
| Entropy Cost | 0.01 |
| Reward Scaling | 10.0 |
| Normalise Observations | True |
| Episode Length | 1,000 |

## D.2. Training details

**BridgeWorld.**   We evaluate two fixed tabular policies that deterministically attempt the left versus the right bridge. Each policy moves laterally to align with the chosen bridge and then repeatedly moves forward while on the bridge.

**Continuous control.**   We train task-conditioned policies with PPO (Schulman et al., 2017). Our implementation is based on Brax (Freeman et al., 2021). We use observation normalisation and reward scaling (factor 10). The actor network has 4 hidden layers of width 32 and the value network has 5 hidden layers of width 256, both with Swish activations. PPO hyperparameters are listed in Table 1. Videos of trained policies across tasks are included in the supplementary material.

**Zones.**   We train an LTL-conditioned policy using DeepLTL (Jackermeier & Abate, 2025) with the default hyperparameters reported in that work for *ZoneEnv*. DeepLTL encodes LTL specifications into a sequential representation processed by an RNN that is trained jointly with the policy.

## D.3. Ablation

### D.3.1. ABLATING CONFIDENCE PARAMETERS $\beta$ AND $\delta$

Across domains, we vary the confidence parameters $(\beta, \delta)$ and report the resulting certified safety curves across multiple task and rollout budgets $(n, m)$, depicted in Figures 7–9. The qualitative behaviour matches Section 5: the shape and tightness of the certificate are determined by the interaction between task-level generalisation from $n$ tasks and per-task estimation from $m$ rollouts, as governed by $(\beta, \delta)$. In particular, varying $\beta$ trades off the conservativeness of the per-task lower bounds against the probability that sufficiently many per-task bounds hold simultaneously in the distribution-level lift, while $\delta$ sets the overall reliability target for the final certificate. Across domains and budgets, we find that the simple guideline $\beta \approx \delta/n$ is a stable default that yields near-best certificates in our runs. Increasing the rollout budget $m$ always tightens the per-task lower confidence bounds, which shifts the certified safety curve to the right in $B$ and improves the certified safety level for any fixed performance threshold.

### D.3.2. ABLATING PER-TASK LOWER CONFIDENCE BOUNDS

For the real-valued return domains (*Cheetah* and *Walker*), we compare different constructions of the per-task lower confidence bounds $\tilde{J}^{\beta}_{\mathcal{M}_i}(\pi)$: Hoeffding, DKW-based bounds, and empirical Bernstein (Appendix A), shown in Figures 10 and 11. Across the settings we consider, empirical Bernstein tends to give the tightest bounds at larger rollout budgets, while DKW is often most competitive at smaller $m$. Hoeffding depends only on the known range and does not exploit the empirical distribution beyond the mean. In our experiments, Hoeffding is dominated by DKW. DKW leverages the empirical CDF shape and can be sharp even when $m$ is small, and empirical Bernstein adapts to the empirical variance. The variance adaptation becomes more effective with larger $m$ as the sample variance stabilises, which can be substantially tighter than worst-case range-based calibration when rollout outcomes have low variability.

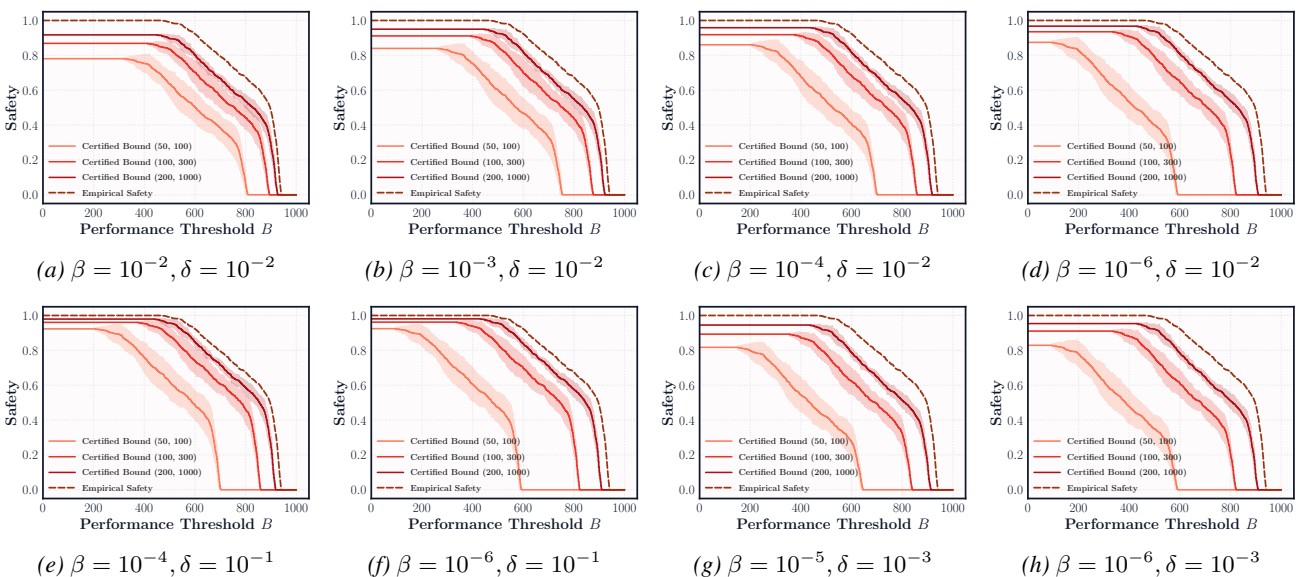

*Figure 7.* Cheetah ablation across $(\beta, \delta)$ settings.

*Table 2.* Runtime for certificate computation per performance bound $B$, obtained from all the computed guarantees over the ablations. We report average runtime and standard deviation over the 10 runs.

| Environment | $n$ | $m$ | Mean (s) | Std (s) |
|---|---|---|---|---|
| cheetah | 50 | 100 | 0.0066 | 0.0089 |
| cheetah | 100 | 300 | 0.0081 | 0.0046 |
| cheetah | 200 | 1000 | 0.0125 | 0.0023 |
| walker | 50 | 100 | 0.0055 | 0.0074 |
| walker | 100 | 300 | 0.0066 | 0.0045 |
| walker | 200 | 1000 | 0.0112 | 0.0027 |
| ZoneEnv | 50 | 100 | 0.0052 | 0.0074 |
| ZoneEnv | 100 | 300 | 0.0088 | 0.0074 |
| ZoneEnv | 200 | 1000 | 0.0157 | 0.0073 |

For the binary-metric domains (*Zones* and *BridgeWorld*), we do not include an analogous ablation, since the Clopper–Pearson interval is the canonical exact finite-sample bound for Bernoulli outcomes and provides guaranteed coverage without asymptotic approximations.

### D.3.3. RUNTIME FOR GUARANTEE COMPUTATION

Table 2 reports the mean and standard deviation of certification time across seeds, tasks, and sampling settings. Across all environments, certificate computation is fast and lightweight. The modest runtime increase with larger $n$ is expected: the search over admissible $K$ grows with the number of sampled tasks.

### D.3.4. EMPIRICAL PERFORMANCE DISTRIBUTION

For completeness, Figure 12 reports the empirical distribution of the policy's per-task performance under the task distribution. This provides a direct view of the induced performance random variable $J_{\mathcal{M}}(\pi)$ and, in particular, an approximation of its CDF. Our safety level $S_{\mathcal{D}}^{\pi}(B) = \Pr_{\mathcal{M} \sim \mathcal{D}}[J_{\mathcal{M}}(\pi) \geq B]$ is exactly the complementary CDF at threshold $B$, and the certified curves in Figures 7–11 are lower bounds on this quantity. The shape of the safety curves therefore mirrors the mass and tails of the empirical performance distribution: thresholds $B$ that lie in high-density regions lead to sharper drops in $S_{\mathcal{D}}^{\pi}(B)$, while flat regions correspond to ranges where little probability mass is present.

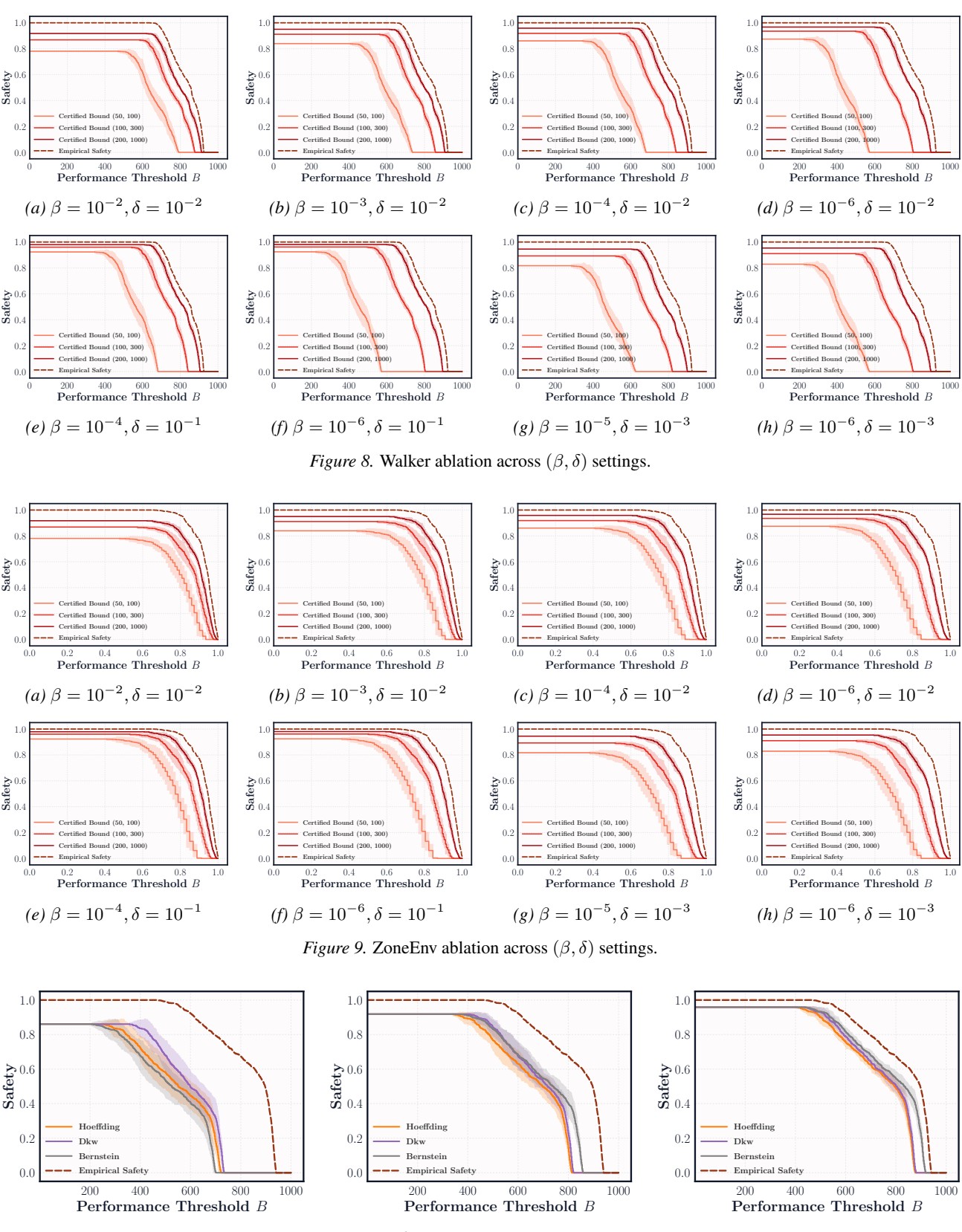

*Figure 8.* Walker ablation across $(\beta, \delta)$ settings.

*Figure 9.* ZoneEnv ablation across $(\beta, \delta)$ settings.

*Figure 10.* Cheetah bounds comparison (Hoeffding vs DKW vs Bernstein) with $\beta = 10^{-4}, \delta = 10^{-2}$.

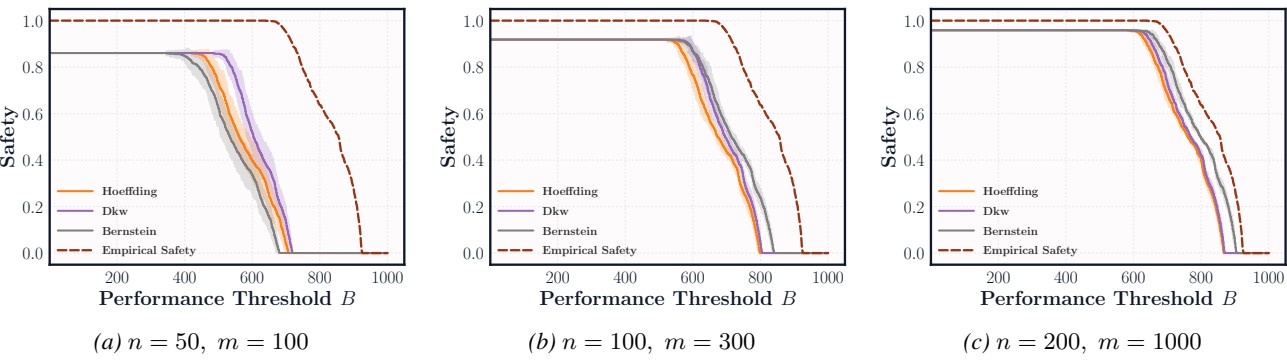

*(a)* $n = 50,\ m = 100$  *(b)* $n = 100,\ m = 300$  *(c)* $n = 200,\ m = 1000$

*Figure 11.* Walker bounds comparison (Hoeffding vs DKW vs Bernstein) with $\beta = 10^{-4}, \delta = 10^{-2}$.

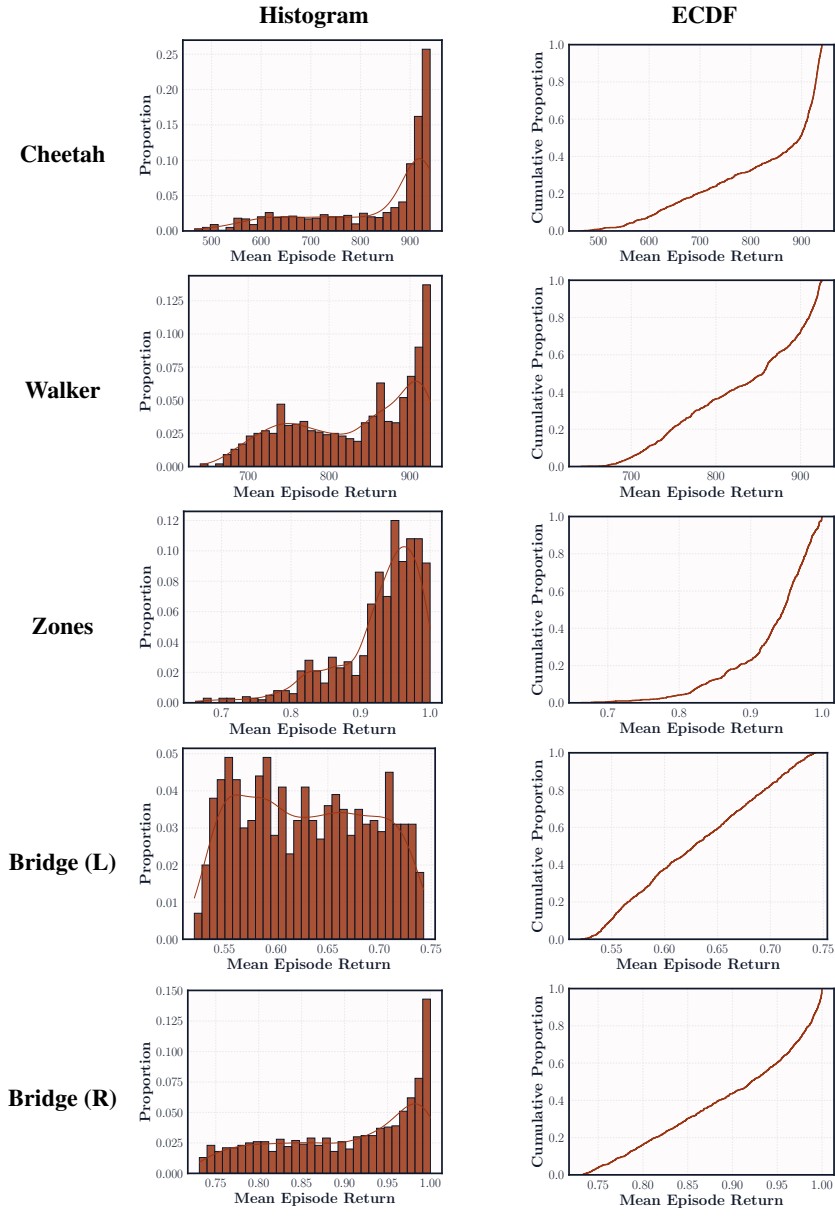

*Figure 12.* Performance distributions across environments. Left column shows histograms of mean episode returns; right column shows empirical cumulative distribution functions (ECDFs).

