# OpenReview forum: "Probabilistic Performance Guarantees for Multi-Task Reinforcement Learning"
_ICML.cc/2026/Conference — ICML 2026 regular_

### Official Review · Reviewer_byDD · 2026-02-24

**Soundness:** 3
**Presentation:** 2
**Significance:** 3
**Originality:** 2
**Overall Recommendation:** 4
**Confidence:** 3

**Summary:**

This paper proposes an approach for obtaining high-confidence performance guarantees in multi-task reinforcement learning. The framework is broadly applicable, independent of the specific training algorithm or any parametric assumptions on the underlying task distribution. Overall, the paper is clearly motivated and provides a principled approach to safety certification beyond fixed single-task evaluation.

**Compliance With Llm Reviewing Policy:**

Affirmed.

**Final Justification:**

Thank the authors for their detailed response. I would like to keep my score.

**Key Questions For Authors:**

1. What precisely is meant by “unseen tasks”? Does this refer to tasks not used during training, or tasks not sampled during the certification procedure? Should these “unseen tasks” belong to the same underlying task distribution (with positive probability mass or density)? More generally, since the methodology focuses on learning probabilistic guarantees at the distribution level rather than for a specific unseen task, it would be helpful to clarify this distinction explicitly.

2. I am somewhat unclear about the primary output of the probabilistic performance guarantee: is it $1-\varepsilon$, $S_D^\pi(B)$, or $B$? What motivates the formulation of the guarantee, rather than directly providing a lower bound on the performance metric at a prescribed significance level? A comparison between the proposed safety certificate and related statistical concepts (e.g., confidence bound or Value-at-Risk) would help clarify its interpretation and positioning.

3. In Theorem 5.1, can the authors provide a more explicit quantification of $\varepsilon$, such as an upper bound or convergence rate, to clarify how it depends on key factors (e.g., the number of sampled tasks, the number of rollouts per task)? Additionally, how does the theoretical result reconcile with the empirical trends.

4. It would be helpful to include a clear algorithm or procedural description specifying the inputs and outputs of the method. In particular, which parameters are predetermined and which quantities are estimated or computed from the sampled tasks and rollouts?

Minor.
L230: $m_i$ or $m$?
L268: $\tilde J_i$ or $\tilde J_{\mathcal{M}_i}(\pi)$?

**Limitations:**

yes

**Strengths And Weaknesses:**

Strengths
1. The resulting certificate is model-free, which does not rely on strong parametric assumptions on task-level performance distributions, making the approach broadly appealing.
2. The theoretical result provides finite-sample guarantees with explicit confidence control over both rollout and task distribution uncertainties.
3. The resulting certification procedure is computationally lightweight.

Weaknesses
1. The motivation of introducing two sources of uncertainty for building a lower bound of the lower bound probability could be better articulated. It would help to compare related statistical objects (e.g., confidence interval/bound, or VaR) and to explain why this choice is preferable for the intended use case.
2. The main theorem presents the result in a somewhat implicit and simplified form. Providing an explicit bound/rate or a corollary that characterizes how the gap between the certified bound and the true safety level scales with key factors (e.g., the number of sampled tasks and the number of rollouts per task) would improve interpretability and help reconcile the theoretical guarantees with the empirical trends.

---

> ### Author Rebuttal · Authors · 2026-03-30
>
> We thank the reviewer for recognising the broad appeal of our model-free approach, our computationally lightweight certification procedure, and the value of our finite-sample guarantees. We appreciate the opportunity to clarify the problem setting and the interpretation of our theoretical results.
>
> ---
>
> **W1 & Q2: Two Sources of Uncertainty, and Comparison to other Statistical Objects**
>
> To clarify, we are not *introducing* two sources of uncertainty. These two sources are inherently present in any realistic MTRL or meta-RL setting:
>
> - We must generalise from a finite sample of tasks to an unknown distribution over potentially uncountably many tasks (e.g., all possible weather conditions).
>
> - Within each sampled task, we cannot compute the policy's exact performance analytically, but must estimate it from a finite number of rollouts.
>
> Existing statistical objects (like standard confidence intervals) address only a single layer of uncertainty, i.e., either task-sampling (assuming exact per-task performance is known) or rollout-sampling (assuming a single, fixed task). Neither can soundly account for both layers simultaneously, which is why our novel method is necessary. Section 5.3 further discusses the difference to roll-out uncertainty where only a single layer of uncertainty is involved.
>
> Regarding the primary output and positioning: Our method essentially provides a high-confidence bound on the Value-at-Risk (VaR), soundly accounting for the two inherently present layers of uncertainty described above.
>
> - **Inputs:** The user provides a desired performance threshold $B$ (the Value at Risk).
> - **True Quantity:** $S^\pi(B)$ is the true, uncomputable safety level (i.e., the true probability that the policy achieves at least $B$ on a new task).
> - **Primary Output:** The output of our method is $1-\epsilon$. This is a provably sound lower bound on $S^\pi(B)$. We certify that, with high confidence over the data collection, the policy will meet the performance $B$ on a new task with probability at least $1-\epsilon$.
>
> We will explicitly draw this connection to VaR in the final version to aid interpretability.
>
> ---
>
> **W2 & Q3: Explicit Convergence Rates**
>
> Our result in Theorem 5.1 is a fully explicit finite-sample bound. For any given data regime ($n$ tasks, $m$ rollouts), our result directly yields a valid high-confidence bound on the safety level.
>
> The overall gap between our certified bound $(1-\epsilon)$ and the true safety level $S^\pi(B)$ can be bounded as a corollary by composing the error rates from the two layers of uncertainty:
>
> - **Rollout-Level Estimation (Scaling with $m$):** Within each task, we estimate performance from $m$ rollouts to compute a lower confidence bound valid with probability $1-\beta$. Deriving the lower confidence bounds using Hoeffding’s inequality, as done in the paper, each estimation error between the true per-task performance and our lower bound scales as $\mathcal{O}\left( \sqrt{\frac{1}{m} \log \frac{1}{\beta}} \right)$.
>
> - **Task-Level Generalization (Scaling with $n$):** Generalising from the $n$ sampled tasks to the unknown underlying distribution introduces a finite-sample penalty. By applying a standard Chernoff bound to the binomial tail in Equation 6, this task-level generalisation gap scales as $\mathcal{O}\left( \frac{1}{n} \log \frac{1}{\delta} \right)$.
>
> Under standard regularity conditions the overall gap is bounded by the sum of these two penalties:
>
> $$\text{Gap} \in \mathcal{O}\left( \frac{1}{n} \log \frac{1}{\delta} + \sqrt{\frac{1}{m} \log \frac{1}{\beta}} \right)$$
>
> These theoretical rates reconcile with our empirical trends. Our empirical ablations systematically isolate and demonstrate how the certified bound $1-\epsilon$ tightens in practice as $n$, $m$ scale for different confidence values $\beta$ and $\delta$.
>
> We will add a corollary for the rates alongside these empirical analyses to provide an interpretable picture.
>
> ---
>
> **Q1: What is meant by "unseen tasks"?**
>
> By "unseen tasks" we mean a newly sampled task drawn i.i.d. from the underlying, unknown task distribution. The guarantee bounds the probability of failure across the entire task distribution from which the certification tasks were drawn. (Note: Extending this framework to account for out-of-distribution tasks, such as bounding performance under a quantified sim-to-real distribution shift, is an exciting avenue for future work we are pursuing).
>
> ---
>
> **Q4: Algorithm/Procedural Description**
>
> We will add a standalone algorithm description in the final version clearly detailing:
>
> - **Inputs:** Sampled rollouts from $n$ tasks, user-chosen performance threshold $B$, confidence parameters $\beta$ (for rollouts) and $\delta$ (overall).
>
> - **Computation (internal, automatic):** The construction of the statistical per-task bounds $\tilde{J}$, the counting of violated bounds $k(B)$, and the bisection to solve Eq. 6 for $\epsilon$.
>
> - **Output:** The certified safety bound $1-\epsilon$.

---

> > ### Author Rebuttal · Reviewer_byDD · 2026-04-03
> >
> > Thank the authors for their detailed response. I would like to keep my score.

---

### Official Review · Reviewer_EyTn · 2026-03-13

**Soundness:** 4
**Presentation:** 4
**Significance:** 4
**Originality:** 3
**Overall Recommendation:** 5
**Confidence:** 3

**Summary:**

The paper works on the multi-task reinforcement learning setting, where the tasks are drawn from a distribution. The goal is to derive algorithm-agnostic high-probability lower bounds on the performance on an unseen task given a set of rollouts from sampled tasks. The performance is measured w.r.t. binary or bounded real-valued trajectory metrics that are assumed to be monotone. The monotonicity assumption is because, for infinite horizon settings, the finite trajectories are required to be lower bound the full trajectory. The main contribution is obtaining a probabilistic lower bound on the unseen task performances given probabilistic lower bounds on the observed task performances. The theoretical result is verified via experiments.

**Compliance With Llm Reviewing Policy:**

Affirmed.

**Final Justification:**

The authors addressed my concerns and reinforced my positive assessment.

**Key Questions For Authors:**

1. Could you discuss the differences between your paper and [1] ? Could you give examples of cases where [1] cannot provide a certificate and your method can (and vice versa)?
2. Could you comment on the scalability of your method? In the related works section, it is noted that the complexity algorithm from [1] depends on the complexity of a constructed model. How does your algorithm behave with the task complexity? Does the complexity also increase due to the increased difficulty of the lower bound estimation in more complex/higher-dimensional environments?
3. Could you elaborate on how your method and [1] depend on task structure and task distribution? When do you expect your method to perform well? How does it differ from [1]?

[1] Schnitzer, Y., Abate, A., and Parker, D. Certifiably robust policies for uncertain parametric environments. In TACAS (3), volume 15698 of Lecture Notes in Computer Science, pp. 63–83. Springer, 2025.

**Limitations:**

yes

**Strengths And Weaknesses:**

1. The paper is well-written. The problem is clearly defined, and the proposed solution is explained in detail along with the reasoning behind the design choices. I think the paper is an enjoyable read.
2. The experiments are quite exhaustive.
3. The proposed method is easy-to-understand, fast and effective as shown by the experiments.
4. My exposure to the area of this paper is narrow. Therefore, I cannot comment on the novelty. Perhaps the authors and other reviewers can clarify this better.
5. The theoretical contributions are limited. The result is an application of a theorem from [1].

[1] Schnitzer, Y., Abate, A., and Parker, D. Certifiably robust policies for uncertain parametric environments. In TACAS (3), volume 15698 of Lecture Notes in Computer Science, pp. 63–83. Springer, 2025.

---

> ### Author Rebuttal · Authors · 2026-03-30
>
> We thank the reviewer for their constructive review and encouraging feedback. We are pleased that they found the paper to be an enjoyable read with clearly defined problems, and that they highlighted our exhaustive experiments and the fast, effective nature of our proposed method.
>
> ---
> **Q1 & Q3: Differences from [1], Applicability, and Task Structure**
>
> The core difference between the [1] and this work lies in how the per-task performance lower bounds are derived, which drastically changes the scope, applicability, and data-efficiency of the methods.
>
> In [1], the method explicitly constructs Interval MDPs (IMDPs) to overapproximate the environmental uncertainty. This inherently restricts the approach to finite-state models with a known transition structure. In contrast, we bypass model construction entirely by estimating performance directly from trajectory returns.
>
> **Applicability:** Because of its reliance on finite-state IMDPs, [1] cannot provide certificates for continuous-control domains (like the Cheetah or Walker). However, it can certify objectives without requiring the monotonicity assumption. In contrast, estimating performance directly from finite rollouts requires us to assume monotonicity for completely arbitrary objectives (which can be relaxed for concrete objectives like discounted returns, as discussed in our response to Reviewer 1). However, for domains where these conditions are met, our method extends applicability far beyond [1] into continuous and structurally unknown MDPs where [1] is fundamentally inapplicable.
>
> **Task Distribution and Structure:** Both methods scale similarly well with respect to the task distribution. However, they differ in how they handle task structure. Since [1] must build a formal IMDP overapproximation, its sample complexity does not scale well with the size of the state space. The method in [1] must divide the user's overall confidence budget across every unknown transition in the model. As the state and action spaces grow, this confidence-splitting forces the individual transition intervals to become highly conservative. Hence, tightening these bounds requires an impractically massive number of rollouts.
>
> Our method bypasses this bottleneck. By estimating performance directly from trajectory returns rather than approximating transition dynamics, our certificates do not degrade with state-space dimension. The tightness of our bounds depends only on the sample size, avoiding the curse of dimensionality. Due to this structural decoupling, we are able to focus on complex, continuous domains where the method in [1] is fundamentally inapplicable.
>
> **We conducted a comparison** on the finite-state BridgeWorld from the paper (Figure 4a) that analyses the tightness of the guarantees from the same number of tasks and roll outs. We also used a larger version of the benchmark to demonstrate how the approach in [1] degrades with growing domain complexity while our approach remains tight guarantees. The comparison is available here:
>
> https://tinyurl.com/3f4f93hs
>
> We will add this comparison and a more detailed discussion on the differences to the final version.
>
> ---
> **Q2: Scalability**
>
> The new algorithm scales exceptionally well compared to [1]. In [1], the algorithm's complexity is heavily tied to the curse of dimensionality. First, it requires formal model checking (robust value iteration) over the constructed IMDPs, which becomes computationally prohibitive as the state space grows. Second, the user's confidence budget must be split across every single transition in the MDP. In complex environments, the local uncertainties compound during value iteration, resulting in conservative bounds that require ten to hundreds of thousands of rollouts to tighten.
>
> Conversely, our algorithm's complexity is decoupled from the dimensionality of the state space. The lower bound estimation only cares about the scalar values of the rollouts. Hence, our complexity and computation time do not increase with task dimensionality.
>
> ---
> **Theoretical Contribution:**
>
> We appreciate that we build upon the scenario-based theorem in [1]. However, we point out that our contribution is not merely a direct application. Instead, we adapt the theorem to integrate per-task lower confidence bounds built strictly from raw trajectory rollouts, bypassing the limiting IMDP overapproximations required by [1].
>
> Further, we adapt the framework to allow for a user-friendly selection of a safety bound $B$. The original theorem in [1] only allows for a pre-defined discarding of a fixed number of observed sample environments, and is restricted to bounding the risk relative to the minimum or a pre-defined $K$-th order statistic of the sampled environments. We adapt this to allow for the user-friendly certification of an arbitrary, pre-specified safety bound $B$, creating a more flexible and applicable framework.
>
> We will highlight these distinctions more explicitly in the final version of the paper.

---

> > ### Author Rebuttal · Reviewer_EyTn · 2026-04-04
> >
> > I thank the authors for their response. My concerns are resolved and I increase my score accordingly.

---

### Official Review · Reviewer_ZLAM · 2026-03-14

**Soundness:** 3
**Presentation:** 4
**Significance:** 3
**Originality:** 3
**Overall Recommendation:** 5
**Confidence:** 4

**Summary:**

This paper provides theoretical performance guarantees of multi-task policies. Multi-task RL presents a twofold challenge for provable guarantees: (1) The return within an MDP is approximated using a finite number of rollouts; (2) Only a finite number of MDPs are sampled during policy training, adding another layer of uncertainty for estimating $\mathbb{E}_{m\sim D}[\mathbb{E}_{\tau\sim m^{\pi}}[G(\tau)]]$.
This study combines these two types of uncertainty and establishes a PAC-style performance guarantee for a policy that is deployed on a newly sampled task. Experiments compare the theory-based confidence bound with empirical ones for several SOTA algorithms in meta-RL, highlighting its soundness and tightness regardless of problem dimensionality.

**Compliance With Llm Reviewing Policy:**

Affirmed.

**Final Justification:**

The authors confirmed my posititve assessment of their work.

**Key Questions For Authors:**

- Could the confidence-bound benchmark how different meta-RL algorithms satisfy safety constraints and accordingly measure how reliable each one is?

- I understand how the monotonicity assumption is crucial for lower-bounding the return. Could the authors think of a way to relax it?

**Limitations:**

The authors should better highlight the monotonicity assumption as a limitation.

**Strengths And Weaknesses:**

**Strengths**

- The paper is excellently written. I appreciated the authors' efforts in providing comprehensive explanations and pedagogical examples to help the reader grasp the intuition and better appreciate the value of their work.

- The paper narrows an ever-growing gap between empirical studies and theoretical contributions in the meta-RL literature.

- It addresses an important challenge in safety-critical tasks, where empirical success on simulated tasks may not be sufficient for policy deployment. The confidence bound provably measures the probability that a policy will be successful upon deployment.

**Weaknesses**

- The monotonicity assumption on the infinite-horizon return sounds restrictive, as it sets aside tasks with a chain reaction of failures, for example.

- Minor: l. 150 should be $h\in\mathbb{N}$

---

> ### Author Rebuttal · Authors · 2026-03-30
>
> We thank the reviewer for their encouraging feedback and for recognising that our paper "narrows an ever-growing gap between empirical studies and theoretical contributions" in the meta-RL literature. We are glad the reviewer appreciated our efforts to provide comprehensive explanations, and that they highlighted the significance of our confidence bounds for safely deploying policies in real-world, safety-critical tasks.
>
> ---
>
> **Q1: Could the confidence-bound benchmark how different meta-RL algorithms satisfy safety constraints?**
>
> Yes, absolutely. This is a primary use case we envision for our framework. Because our certification method is algorithm-agnostic, it can naturally serve as a standardised and principled benchmarking tool to evaluate and compare the safety and reliability of different MTRL / meta-RL algorithms. We briefly demonstrated this capability in our BridgeWorld experiment, where we used the bound to quantitatively compare the certified safety of two distinct policies. We agree that adopting this as a principled evaluation metric for MTRL algorithms would be highly valuable, and we will emphasise this benchmarking application more prominently in the final version.
>
> ---
>
> **Q2 & Limitations: The monotonicity assumption for infinite-horizon sounds restrictive / Could you think of a way to relax it?**
>
> For completely arbitrary trajectory metrics and infinite horizon, monotonicity is a minimal required assumption. Without it, an environment could theoretically trigger an unbounded "chain reaction of failures" at $t \to \infty$, which would make it impossible to establish a lower bound from a finite prefix.
>
> However, for concrete, standard RL objectives like the infinite-horizon discounted return, this assumption is not restrictive and is easily relaxed. If an environment has negative rewards bounded by some $R_{min} < 0$, we do not have monotonicity but we can handle it in two ways without changing the underlying problem:
>
> 1. **Worst-Case Geometric Limit:** We can redefine our observable finite-prefix statistic to account for the maximum possible future loss. We can define it as: $\mathcal{G}^*(\tau[..h]) = \sum_{t=0}^{h} \gamma^t r_t + \frac{\gamma^{h+1}}{1-\gamma} R_{min}$ (note that when $R_{min}$ is negative, this is deducing the maximum future loss). This adjusted statistic converges to the true return monotonically from below, satisfying our framework's requirement.
>
> 2. **Reward Shifting:** Alternatively, we can simply shift all rewards by a constant to make them non-negative ($r_t’ = r_t - R_{min} \ge 0$, again when $R_{min}$ is negative, this is increasing the rewards). As established by Ng, Harada, and Russell (1999) in their work on reward shaping, applying a constant shift to all rewards in a discounted MDP preserves the relative ordering of all policies. To apply our certificate, one simply shifts the user's performance threshold $B$ accordingly by the constant $R_{min}/(1-\gamma)$, which in turn implies a threshold of $B$ on the original reward structure.
>
> We will explicitly highlight the monotonicity assumption as a limitation for arbitrary infinite-horizon metrics in the final version, and we will add a dedicated discussion explaining how it can be relaxed for standard RL objectives.

---

> > ### Author Rebuttal · Reviewer_ZLAM · 2026-04-03
> >
> > The authors fully answered my questions. I encourage acceptance.

---

### Decision · Program_Chairs · 2026-04-30

**Decision:**

Accept (regular)

**Comment:**

The paper presents a certification method for multi-task reinforcement learning
that yields high-confidence safety guarantees for a fixed policy on a new task
drawn from an unknown task distribution. The core contribution is composing
rollout-based per-task lower confidence bounds with a task-level generalisation
argument, producing an interpretable certificate of the form


$$
\Pr\!\left[S^\pi_\mathcal{D}(B) \geq 1 - \varepsilon\right] \geq 1 - \delta
$$


for a user-chosen performance threshold $B$ and confidence level $1 - \delta$.

Reviews were positive, with two accepts and one weak accept. The
paper is well-written, the problem is well-motivated, and the experimental
evaluation is thorough across diverse environments including high-dimensional
continuous control.


The main theoretical novelty question raised by Reviewer EyTn, namely the
relationship to Schnitzer et al. (2025), was addressed adequately. The authors
correctly explain that the present paper adapts the scenario-based framework to
a rollout-based setting where per-task performances are not observed exactly but
only bounded probabilistically which extends applicability far beyond the finite-state IMDP setting of
Schnitzer et al. The proof in Appendix C makes this adaptation explicit and the
distinction is real and non-trivial.

The monotonicity assumption is a genuine restriction for arbitrary
infinite-horizon metrics, but the authors' discussion of how it can be relaxed
for standard RL objectives, such as discounted return with negative rewards
via worst-case geometric truncation or reward shifting, is convincing. This
discussion should appear prominently in the final version as promised, rather
than only in the rebuttal.